# TMIGD2 is an orchestrator and therapeutic target on human acute myeloid leukemia stem cells

Hao Wang[1], R. Alejandro Sica[2], Gurbakhash Kaur[3], Phillip M. Galbo Jr [1,4], Zhixin Jing [5], Christopher D. Nishimura[1], Xiaoxin Ren[1], Ankit Tanwar[1], Bijan Etemad-Gilbertson[6], Britta Will[2,7], Deyou Zheng [4,8], David Fooksman[5] & Xingxing Zang [1,2,9,10] ✉

Acute myeloid leukemia (AML) is initiated and sustained by a hierarchy of leukemia stem cells (LSCs), and elimination of this cell population is required for curative therapies. Here we show that transmembrane and immunoglobulin domain containing 2 (TMIGD2), a recently discovered co-stimulatory immune receptor, is aberrantly expressed by human AML cells, and can be used to identify and enrich functional LSCs. We demonstrate that TMIGD2 is required for the development and maintenance of AML and self-renewal of LSCs but is not essential for normal hematopoiesis. Mechanistically, TMIGD2 promotes proliferation, blocks myeloid differentiation and increases cell-cycle of AML cells via an ERK1/2-p90RSK-CREB signaling axis. Targeting TMIGD2 signaling with anti-TMIGD2 monoclonal antibodies attenuates LSC self-renewal and reduces leukemia burden in AML patient-derived xenograft models but has negligible effect on normal hematopoietic stem/progenitor cells. Thus, our studies reveal the function of TMIGD2 in LSCs and provide a promising therapeutic strategy for AML.

Acute myeloid leukemia (AML) is an aggressive clonal hematopoietic disorder characterized by a block in myeloid differentiation and thus an accumulation of uncontrolled immature myeloid cells in the peripheral blood and bone marrow (BM). Despite rapid advances over the past 10-15 years having improved our understanding of the molecular and phenotypic diversity of AML, poor to moderate anti-leukemic response or relapse following short remission still remains a major challenge to the treatment of AML[1,2]. Therapy-naïve and adaptively acquired leukemia stem cells (LSCs) possess several stem cell properties including self-renewal, cell-cycle quiescence and chemoresistance. LSCs have long been described as sources of refractory disease to standard treatments, such as conventional chemotherapy, hypomethylating agents, and hematopoietic stem cell transplantation, resulting in an approximately 75% relapse rate and 30% 5-year overall survival rate in older patients[1,3–6]. Moreover, due to the co-existing and evolution of heterogenous LSC subclones within individual patient upon targeted treatment, the efficacy of therapy targeting one molecule expressed only in a subset of AML might be limited[7–10]. Therefore, the identification of therapeutic targets broadly and consistently expressed on LSCs to maximize the clinical potential of AML treatment as well as precisely predict patient outcome is an unmet clinical need[6,11,12].

[1]Department of Microbiology & Immunology, Albert Einstein College of Medicine, Bronx, NY 10461, USA. [2]Department of Medicine, Albert Einstein College of Medicine, Bronx, NY 10461, USA. [3]Department of Internal Medicine, UT Southwestern Medical Center, Dallas, TX 75390, USA. [4]Department of Genetics, Albert Einstein College of Medicine, Bronx, NY 10461, USA. [5]Department of Pathology, Albert Einstein College of Medicine, Bronx, NY 10461, USA. [6]NextPoint Therapeutics, Inc, Cambridge, MA 02142, USA. [7]Department of Cell Biology, Albert Einstein College of Medicine, Bronx, NY 10461, USA. [8]Departments of Neurology and Neuroscience, Albert Einstein College of Medicine, Bronx, NY 10461, USA. [9]Department of Urology, Albert Einstein College of Medicine, Bronx, NY 10461, USA. [10]Department of Oncology, Albert Einstein College of Medicine, Bronx, NY 10461, USA. ✉e-mail: xingxing.zang@einsteinmed.edu

We and others previously identified the transmembrane and immunoglobulin domain containing 2 (TMIGD2, also known as CD28H or IGPR1), a new immune molecule belonging to the CD28 family, as a co-stimulatory receptor for the B7 family member HERV-H LTR-associating 2 (HHLA2)[13–16]. TMIGD2 is a transmembrane protein mainly expressed on T cells and natural killer (NK) cells[16,17]. HHLA2 engagement co-stimulates human T and NK cells through TMIGD2, resulting in cell proliferation and cytokine production[16,18]. Importantly, TMIGD2 and HHLA2 have orthologs in humans and monkeys but not in laboratory mice or rats[14]. Despite such an established understanding of TMIGD2 in T and NK cells, its role in normal hematopoietic stem/progenitor cells (HSPCs) and LSCs is unknown.

Here, we reported an unexpected function and a new therapy of TMIGD2 on LSCs of human AML. We showed that the expression of TMIGD2 was highly upregulated in CD34+ AML cells, where it could enrich for functional LSCs. Furthermore, TMIGD2 played critical roles in regulating self-renewal and differentiation of LSCs, as well as facilitating leukemogenesis in vivo, but was not functionally required for normal HSPCs. In an effort to develop pharmacological inhibition of the TMIGD2 oncogenic signaling pathway in AML, we generated distinct anti-TMIGD2 mAbs, which provided therapeutic effect in AML patient-derived xenograft (PDX) models of refractory/relapsed AML without derailing normal hematopoiesis, and simultaneously maintained HHLA2/TMIGD2-mediated co-stimulatory signaling on T and NK cells.

## Results

### TMIGD2 is more highly expressed on AML LSCs than on their normal counterparts

To explore whether the HHLA2-TMIGD2 pathway plays a novel role in regulating AML development and mediating immune responses, we analyzed the cell surface expression of HHLA2 and TMIGD2 in 44 AML patient samples by flow cytometry (Fig. 1a and Supplementary Table 1). Strikingly, TMIGD2, but not HHLA2, was selectively enriched in CD45dimSSClowLin-CD34+ cells (Fig. 1b, d). By analyzing a database containing AML samples with normal cells[19], we found that the pattern of HHLA2 and TMIGD2 protein expression was also consistent with the data available at the mRNA level (Supplementary Fig. 1a, b). Furthermore, results from gene correlation analysis indicated a positive association between TMIGD2 and CD34 (Supplementary Fig. 1c). In The Cancer Genome Atlas (TCGA) cohort, TMIGD2 was more prominently expressed in FAB M0-M2 subtypes, acute promyelocytic leukemia (FAB M3), and FAB-M6 acute erythroid leukemia (Supplementary Fig. 1d)[20]. As LSCs mainly reside in CD34+CD38- and CD34+CD38+ compartments[3,21], we further examined those subsets of our AML specimens and found that TMIGD2 was more highly expressed on CD34+CD38- and CD34+CD38+ compartments in comparison with CD45dimSSClow unfractionated blasts (Fig. 1c). Given that CD45RA, CD123, and IL1RAP were previously identified as LSC-specific markers[8,11,22–24], we investigated their co-expression with TMIGD2 within the CD34+ subpopulation. In 95% of 41 AML samples, AML cells from the CD34+TMIGD2+ subset expressed at least one of the CD45RA, CD123 and IL1RAP markers (Supplementary Fig. 1e). Of note, TMIGD2 was largely co-expressed with CD45RA within both CD34+CD38- and CD34+CD38+ subpopulations (Supplementary Fig. 1f), suggesting that TMIGD2 is predominantly expressed on LSCs.

Interestingly, we observed an obvious upregulation of TMIGD2 in CD34+ AML cells compared with CD34+ HSPCs from 4 cord blood units (CBU) as well as 6 normal adult BM (NBM) samples (Fig. 1e and Supplementary Fig. 1g). Given the role of TMIGD2 in immune cells, we also examined TMIGD2 expression on T cells from AML specimens versus healthy donors. As predicted, due to repetitive neoantigen stimulation, TMIGD2 expression on CD3+ T cells from AML samples was downregulated compared with healthy donors (Fig. 1f). Moreover, in individuals with AML, the expression of TMIGD2 was significantly higher in CD34+ AML cells than in CD3+ T cells (Fig. 1g). Kaplan–Meier analysis revealed that the expression of TMIGD2 was a good predictor of AML progression and thus indicative of a worse clinical outcome (Fig. 1h). Notably, AML patients harboring NPM1 mutation, who in many cases often have <1% normal CD34+ cells[11], expressed significantly lower TMIGD2 (Supplementary Table 2), suggesting that TMIGD2 is specifically expressed on the malignant CD34+ cells. Taken together, higher TMIGD2 expression in LSC-residing populations indicates that it may play a critical role in regulating the function of LSCs and AML development.

### Loss of TMIGD2, but not HHLA2, inhibits AML development and promotes myeloid differentiation

To investigate whether TMIGD2 plays a functional role in AML, we intended to identify human AML cell lines expressing TMIGD2. Using the Cancer Cell Line Encyclopedia (CCLE), we evaluated mRNA expression of TMIGD2 across >1000 different malignant cell lines, including 39 AML cell lines[25]. TMIGD2 mRNA was highly expressed in AML and T-cell acute lymphoblastic leukemia cell lines, whereas no other tumor types exhibited substantial TMIGD2 expression (Supplementary Fig. 2a). Human AML cell lines, including HEL, K562, KG-1a, Kasumi-1 and ME-1, were confirmed to express cell surface TMIGD2 protein as analyzed by flow cytometry (Supplementary Fig. 2b). To further evaluate the functions of TMIGD2 and HHLA2 in leukemia cells, we next performed small hairpin RNA (shRNA)-mediated knockdown of TMIGD2 (shTMIGD2) or HHLA2 (shHHLA2) in Kasumi-1, HEL and K562 cell lines (Fig. 2a, b and Supplementary Fig. 2c, d). Loss of TMIGD2, but not HHLA2, led to significant inhibition of leukemia cell growth over the course of 6 days (Fig. 2c and Supplementary Fig. 2e). In addition, the dominant cellular phenotypes that resulted from TMIGD2 knockdown included a substantial induction of apoptosis (Fig. 2d and Supplementary Fig. 2f), a prominent G0/G1 cell-cycle arrest (Fig. 2e, Supplementary Fig. 2g, h), a noticeable increase of differentiation (Fig. 2f, g and Supplementary Fig. 2i–k) and a significant decrease of colony-forming cell (CFC) number (Fig. 2h, Supplementary Fig. 2l) in Kasumi-1, HEL and K562 AML cells. However, these changes were not observed after HHLA2 knockdown (Fig. 2d, g, h and Supplementary Fig. 2f, k). Further analysis via CFC assay revealed that self-renewal capacity of HEL cells remained unchanged after treatment with anti-HHLA2 monoclonal antibodies (mAbs) or HHLA2-human Fc fusion protein (Fig. 2i), indicating that TMIGD2-mediated AML function is HHLA2 independent. Notably, decreases in TMIGD2 expression, both at the surface protein and mRNA levels, were observed when HEL cells were treated with a differentiation-inducing agent, phorbol-12-myristate-13-acetate (PMA), suggesting that the downregulation of TMIGD2 is accompanied by myeloid differentiation (Supplementary Fig. 2m, n).

To investigate the effect of TMIGD2 on transcriptional functions in AML cells, we performed gene expression analyses by RNA-sequencing (RNA-seq) in shCtrl and shTMIGD2 HEL cells (Supplementary Fig. 2o). Gene set enrichment analysis (GSEA) revealed that pathways enriched among differentially expressed genes included mitotic spindle, interferon alpha response, and Runx1 regulated transcription of genes involved in differentiation of HSCs (Supplementary Fig. 2p). Theses pathways reflected differentiation and cell cycling as potential phenotypes associated with TMIGD2 knockdown in HEL cells. Among the genes with significant changes upon TMIGD2 depletion were factors closely associated with cell-cycle (CDKN1A and CCND1) and leukemia stemness (CD93 and IL1RAP; Fig. 2j).

To assess the role of TMIGD2 in leukemogenesis in vivo, we firstly established an AML cell line-derived xenograft model by intravenously transplanting the shCtrl or shTMIGD2 AML cells into sublethally irradiated NSG mice. As shown in Fig. 3a–d and Supplementary Fig. 3a, b, we observed that NSG-recipients of shTMIGD2 HEL and shTMIGD2 K562 cells developed and died of AML significantly slower than did

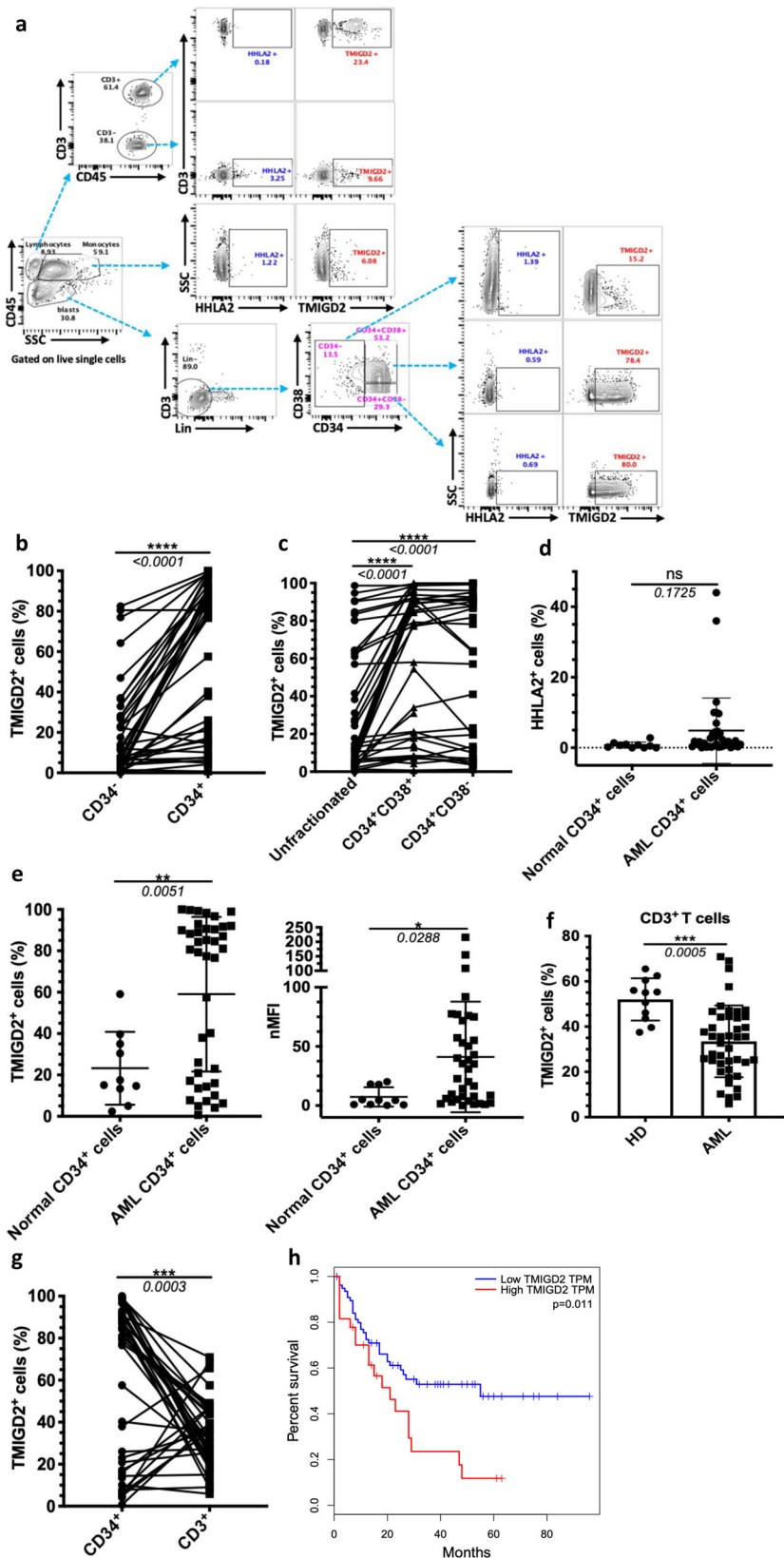

recipients of shCtrl HEL and shCtrl K562 cells. The delayed development of AML in the absence of TMIGD2 was associated with a lower percentage of leukemic cells in the BM and spleen (Fig. 3a, b and Supplementary Fig. 3a), as well as an induction of differentiation (Fig. 3c). Next, we implanted NSG mice with Tet-On shCtrl or Tet-On shTMIGD2 Kasumi-1 cells and treated the NSG-recipients with doxycycline that can induce Tet-On shRNA to silence TMIGD2 (Supplementary Fig. 3c). When induced at day 5 after cell transplantation, the knockdown of TMIGD2 slowed AML development (Fig. 3e, f). Moreover, depletion of TMIGD2 also significantly arrested tumor growth in HEL and K562 subcutaneous xenograft models of AML in NSG mice (Fig. 3g and Supplementary Fig. 3d, e). Overall, these data

**Fig. 1 | TMIGD2 is differentially expressed between stem/progenitor cells and blasts in AML patients. a** Representative flow cytometry gating strategy showing HHLA2 and TMIGD2 expression on monocytes, stem cells (CD34$^+$CD38$^-$), progenitor cells (CD34$^+$CD38$^+$), and T cells (CD3$^+$) of AML patients. Lin, CD14 and CD19. **b** Quantification of TMIGD2 expression on CD34$^+$ and CD34$^-$ AML cells ($n = 44$). **c** Comparison of the expression level of TMIGD2 in stem/progenitor subpopulations and bulk unfractionated blasts ($n = 41$). **d** Quantification of HHLA2 expression in AML CD34$^+$ cells ($n = 36$) and normal CD34$^+$ cells from CBU ($n = 4$) and NBM ($n = 6$) mononuclear cells. **e** Quantification of TMIGD2 expression in CD34$^+$ AML cells ($n = 40$) relative to that in normal CD34$^+$ cells ($n = 10$). Data are shown as percentage of positive (left) and mean fluorescence intensity (MFI, right) relative to the fluorescence minus one (FMO) control. **f** Comparison of TMIGD2 expression on CD3$^+$ T cells from AML patients ($n = 41$) and healthy donors (HD, $n = 12$). **g** Comparison of TMIGD2 expression on CD3$^+$ T cells and CD34$^+$ leukemia cells of human AML samples ($n = 41$). **h** Kaplan–Meier survival analysis using TCGA-AML dataset. The patients were divided into two groups, high group ($n = 79$) > 25% of TMIGD2 level, low group ($n = 79$) < 75% of TMIGD2 level. The p value was calculated by the log-rank test. http://gepia2.cancer-pku.cn/#survival. Mean ± SD values are shown for Fig. 1d–f. Non-significant (ns), $*p < 0.05$, $**p < 0.01$, $***p < 0.001$, and $****p < 0.0001$ by two-tailed Student's $t$ test (**d–f**) or paired Student's $t$ test (**b, c, g**). Source data are provided in the Source Data file.

strongly implicate the importance of TMIGD2 for AML development and maintenance.

We next used intravital two-photon microscopy to study the behavioral response of AML cells in the BM upon TMIGD2 depletion. By intravenous injection of mixed cells at different ratios, we found that shCtrl-tdTomato HEL cells became the predominant cells within the tibia BM (Fig. 3h, i), suggesting that TMIGD2 is essential for the expansion of leukemic cells in the BM. Additionally, time-lapse videos of NSG-recipients injected concomitantly with shCtrl-tdTomato and shTMIGD2-GFP HEL cells revealed that leukemic cell motility was significantly decreased after TMIGD2 knockdown (Supplementary Fig. 3f), and videos also showed more shCtrl-tdTomato cells recirculating within blood sinusoid spaces (Supplementary Fig. 3g). To confirm these results, leukemic cells within the BM vascular niche of recipient NSG mice were flash-labeled by intravenous administration of fluorophore-conjugated antibody[26]. As expected, a higher frequency of labeled shCtrl-tdTomato leukemic cells were observed (Supplementary Fig. 3h). These data, together with the presence of more shCtrl than shTMIGD2 leukemic cells in the spleen of NSG-recipients (Supplementary Fig. 3a), indicate that TMIGD2 contributes to dissemination of leukemic cells.

## TMIGD2$^+$ AML cells are enriched in LSC activity

In most human AML, LSCs reside predominantly within the CD34$^+$ subpopulation of AML blasts, among which they are highly enriched based on CD38 negativity and expression of other previously identified surface markers[3,6,12]. Considering that TMIGD2 was frequently expressed in CD34$^+$CD38$^-$ and CD34$^+$CD38$^+$ cell fractions, we hypothesized that the expression of TMIGD2 could define subset of leukemic cells with distinct leukemogenesis properties. To test this hypothesis, we used fluorescence-activated cell sorting (FACS) to fractionate primary human AML specimens into CD34$^+$CD38$^-$TMIGD2$^+$ and CD34$^+$CD38$^-$TMIGD2$^-$ subpopulations, or CD34$^+$CD38$^+$TMIGD2$^+$ and CD34$^+$CD38$^+$TMIGD2$^-$ subpopulations in samples that CD34$^+$CD38$^-$ subset was absent (Supplementary Table 3). CFC assays were then performed (Fig. 4a). Clonogenic ability was dramatically higher in the TMIGD2$^+$ fraction compared with the TMIGD2$^-$ fraction of CD34$^+$CD38$^-$/CD34$^+$CD38$^+$ cells upon serial replating, suggesting a reduction in LSC frequency and self-renewal in CD34$^+$CD38$^-$ (or CD34$^+$CD38$^+$) TMIGD2$^-$ subpopulation (Fig. 4b, c and Supplementary Fig. 4b). Since we observed TMIGD2 expression in CD34$^-$ cells from a small subset of AML patients whose CD34 expression was limited, we investigated whether TMIGD2 could enrich for functional LSCs in CD34$^-$ AML. CD34$^-$TMIGD2$^+$ AML cells showed enhanced in vitro clonogenicity in comparison with their TMIGD2$^-$ counterpart (Fig. 4b, c). Similar results were obtained when CFC assays were conducted using FACS-purified TMIGD2$^+$ and TMIGD2$^-$ blasts irrespective of CD34 and CD38 expression (Fig. 4c and Supplementary Fig. 4a). To directly enumerate the frequency of LSCs, a limiting dilution xenotransplantation experiment was performed with FACS-purified subsets[27]. Consistent with the results of CFC assay, the CD34$^+$CD38$^-$TMIGD2$^+$, CD34$^-$TMIGD2$^+$ and Lin$^-$TMIGD2$^+$ subpopulations had a substantially higher LSC frequency than corresponding TMIGD2$^-$ subpopulations, with enhanced ability to repopulate NSG mice in xenotransplantation assays (Fig. 4d, Supplementary Fig. 4c–e, and Supplementary Table 3).

To comprehensibly understand the genes and pathways associated with TMIGD2 expression in LSCs, transcriptome-wide RNA-seq was conducted on FACS-purified CD34$^+$TMIGD2$^+$ and CD34$^+$TMIGD2$^-$ subfractions of six primary AML specimens. The RNA-seq analysis revealed that a total of 453 genes were significantly differentially expressed by at least 2-fold (FDR < 0.05; Fig. 4e). Among these differentially expressed genes, the CD34$^+$TMIGD2$^+$ leukemic cells had increased expression of several genes classically associated with LSC signatures and poor prognosis, such as *SCRN1*, *SMIM24*, and *ABCB1*. In contrast, CD34$^+$TMIGD2$^-$ leukemic cells exhibited higher expression of monocyte/granulocyte-associated genes (e.g., *ITGAM*, *CLEC5A*, *FPR1*, *CEBPB*, and *S100A8*; Fig. 4e). Importantly, GSEA showed that the top seven pathways enriched in CD34$^+$TMIGD2$^+$ fraction consisted of E2F targets, MYC targets, and G2M checkpoints (Fig. 4f, g), which was consistent with our findings that CD34$^+$TMIGD2$^+$ cells generated more colonies and induced leukemia much more efficiently than TMIGD2$^-$ counterparts (Fig. 4b–d). By contrast, we found significant enrichment of a set of functionally important signaling pathways in CD34$^+$TMIGD2$^-$ subpopulation, including TNF-α signaling, inflammatory response, apoptosis, and the p53 pathway (Supplementary Fig. 4f, g). Moreover, CD34$^+$TMIGD2$^+$ subpopulation was associated with the established 17-gene stemness (LSC17) and LSC signatures (Fig. 4h), while corresponding TMIGD2$^-$ fraction was correlated with myeloid cell development, hematopoiesis maturation, and downregulation of *HOXA9* and *MEIS1* targets (Supplementary Fig. 4g).

To investigate the self-renewal and differentiation potential of TMIGD2$^+$ and TMIGD2$^-$ subsets in normal HSPCs, a CFC assay and xenotransplantation were performed using FACS-purified CD34$^+$TMIGD2$^+$ and CD34$^+$TMIGD2$^-$ cells from CBU and NBM samples. Both fractions were found to form GEMM (Granulocyte/Erythrocyte/Macrophage/Megakaryocyte), GM (granulocyte/macrophage), and E (erythroid) colonies (Fig. 4i). However, most GM grew out of the CD34$^+$TMIGD2$^+$ fraction, whereas E originated from the CD34$^+$TMIGD2$^-$ fraction (Fig. 4i and Supplementary Fig. 4h, i). In vivo HSPCs activity was indicated by the presence of both hCD45$^+$CD19$^+$ lymphoid and hCD45$^+$CD33$^+$ myeloid cells in the BM of NSG mice 12 weeks after transplantation (Supplementary Fig. 4j). When transplanted at equal cell doses, both lymphoid and myeloid engraftments were detected with a similar frequency in NSG mice engrafted with either TMIGD2$^+$ or TMIGD2$^-$ subsets (Fig. 4j). Taken together, these results indicate that, although normal functional HSPCs reside in both CD34$^+$TMIGD2$^+$ and CD34$^+$TMIGD2$^-$ compartments, LSCs reside predominantly within the CD34$^+$TMIGD2$^+$ compartment in AML.

## TMIGD2 is essential for maintaining primary human AML LSC capability

To define the direct role of TMIGD2 in maintaining LSC function and leukemogenesis, FACS-purified CD34$^+$CD38$^-$ or CD34$^+$CD38$^+$ primary AML cells from patients were transduced with lentivirus expressing shCtrl or shTMIGD2. We next conducted in vitro analyses and

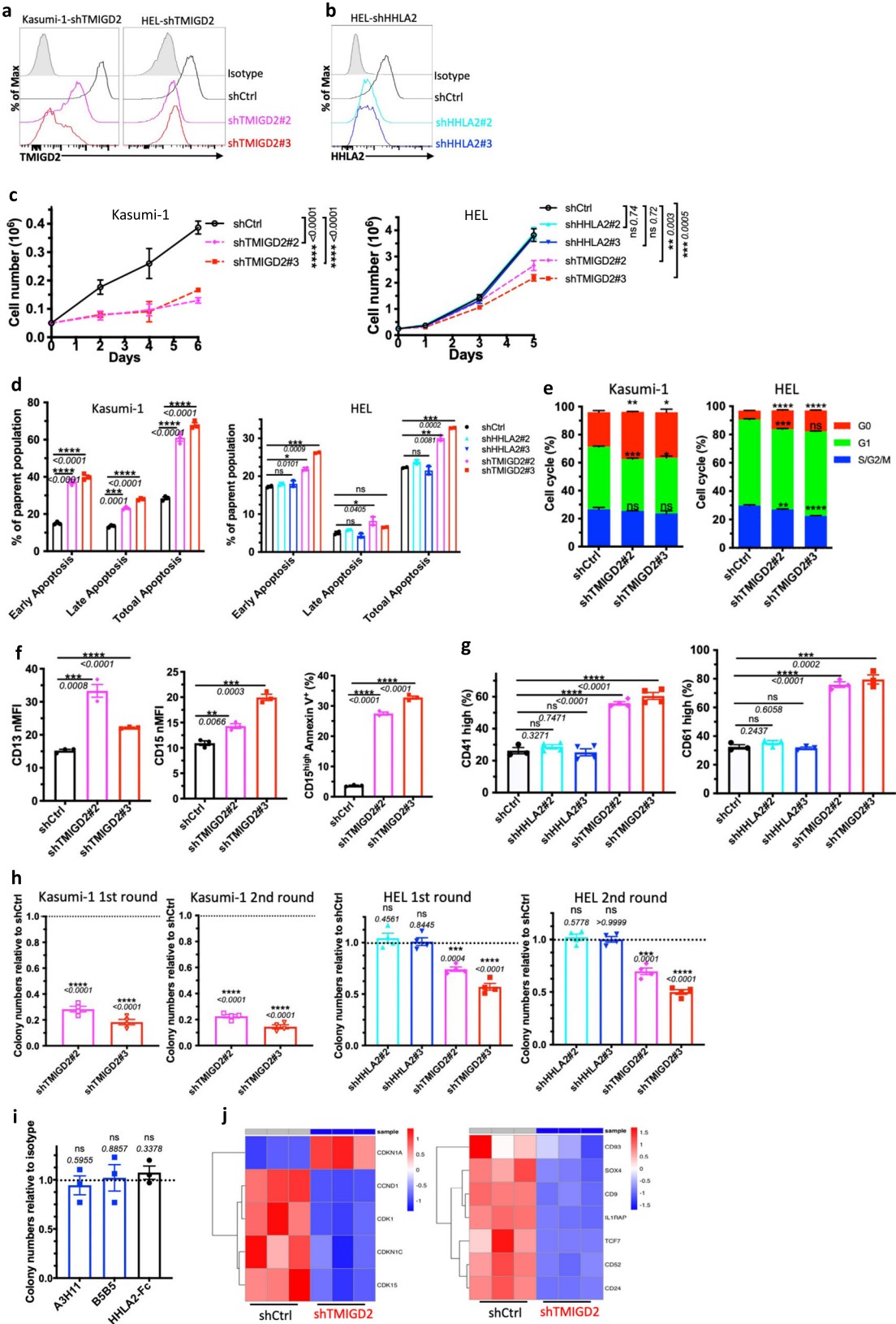

xenotransplantation experiments using the shCtrl and shTMIGD2 primary AML cells (Fig. 5a). TMIGD2 knockdown resulted in increased expression of differentiation marker CD11b and apoptotic marker annexin V in primary AML cells but not in NBM and CBU (Fig. 5b, c and Supplementary Fig. 5a), indicating that TMIGD2 is specifically important for LSCs. In addition, we found that TMIGD2 depletion

significantly decreased the size and number of colonies derived from primary AML cells (Fig. 5d, e), supporting a role for TMIGD2 in promoting proliferation and clonogenic potential of leukemic cells. In agreement with the in vitro data, knockdown of TMIGD2 resulted in significantly delayed human AML progression (Pt#5 and Pt#8), reduced engraftment potential (Pt#24 and Pt#26), and prolonged

**Fig. 2 | TMIGD2 plays a critical role in regulating AML proliferation, self-renewal and differentiation. a** Kasumi-1 (left) and HEL (right) cells were transduced with shRNA targeting TMIGD2 and subjected to flow cytometry analysis. shCtrl, shRNA Control; shTMIGD2, TMIGD2 knockdown. Kasumi-1 is endogenously HHLA2 negative. **b** Knockdown of HHLA2 in HEL cells. shHHLA2, HHLA2 knockdown. **c** Growth curves of Kasumi-1 (left) and HEL (right) cells upon knockdown of HHLA2 or TMIGD2. **d** Quantification of apoptosis analysis in Kasumi-1 (left) and HEL (right) cells after HHLA2 or TMIGD2 knockdown. Early apoptosis, Annexin V$^+$DAPI$^-$; late apoptosis, Annexin V$^+$DAPI$^+$. **e** Quantification of cell-cycle analysis in shCtrl and shTMIGD2 Kasumi-1 (left)/HEL (right) cells. **f** Statistics of surface expression of differentiation markers CD13 (left) and CD15 (middle) on shCtrl and shTMIGD2 Kasumi-1 cells. Right, quantification of CD15$^{high}$Annexin V$^+$ cells in shCtrl and

shTMIGD2 Kasumi-1 cells. **g** Statistics of surface expression of differentiation markers CD41 (left) and CD61 (right) on shCtrl, shHHLA2 and shTMIGD2 HEL cells. **h** First round and second round colony-forming cell (CFC) counts of Kasumi-1 (left) and HEL (right) cells after transduction with indicated lentiviruses. (500 cells/plate). **i** CFC counts of HEL cells after treatment with 50μg/ml anti-HHLA2 mAbs (A3H11 and B5B5) or HHLA2-hFc fusion protein. **j** Heatmap showing differentially expressed genes related to cell-cycle (left) and leukemia stemness (right) in shCtrl versus shTMIGD2 HEL cells. Mean ± SEM values are shown for Fig. 2. ns, *$p < 0.05$, **$p < 0.01$, ***$p < 0.001$, and ****$p < 0.0001$ by two-tailed Student's $t$ test. For **c**–**i**, color dots represent technical replicates. Results are representative of three independent experiments. Source data are provided in the Source Data file.

survival (Pt#5) in AML PDX models (Fig. 5f and Supplementary Fig. 5b, c), indicating that TMIGD2 is essential for the development of AML.

Next, we explored whether TMIGD2 is required for normal hematopoiesis. We knocked down TMIGD2 by shRNAs in human-derived-CD34+ cells and then performed CFC assay and xeno-transplantation. As shown in Fig. 5g, the comparable colony-forming ability was observed by using shCtrl or shTMIGD2 CD34$^+$ cells. Compared with NSG mice engrafted with shCtrl CD34$^+$ cells, mice engrafted with shTMIGD2 CD34$^+$ cells exhibited similar percentages of total hCD45$^+$ cells, hCD19$^+$ lymphoid cells and hCD33$^+$ myeloid cells in the BM (Fig. 5h), suggesting that TMIGD2 deletion has no detectable impact on normal human CD34$^+$ HSPCs.

## TMIGD2 regulates the function of LSC through the ERK1/2-p90RSK-CREB pathway

We next sought to understand how TMIGD2 depletion influences signaling pathways in AML cells. Using a human phospho-kinase array, we first compared signaling protein levels in shCtrl and shTMIGD2 HEL cells. We found that the abundance of phosphorylated CREB (p-CREB) was significantly reduced upon TMIGD2 knockdown (Supplementary Fig. 6a), which was verified by Western blot (Fig. 6a). Considering that CREB can act as either a direct or an indirect substrate for ERK and AKT pathways, which play critical roles in the transmission of proliferative signals from membrane bound receptors in AML cells[28], we next tested whether depletion of TMIGD2 could affect these pathways. Western blot analysis showed that p-ERK1/2 but not p-AKT protein levels were decreased in shTMIGD2 HEL cells compared to control (Fig. 6b). In agreement with these observation in HEL cells, knockdown of TMIGD2 decreased level of p-CREB and p-ERK1/2 in primary CD34$^+$ AML cells but not in CD34$^+$ cells isolated from NBM and CBU (Supplementary Fig. 6b). Furthermore, it has been shown that ribosomal S6 kinases (RSKs) phosphorylate CREB, promoting myeloid cell proliferation and survival through induced expression of Bcl-2 and cyclin-A[29,30]. The Western blot analyses showed that loss of TMIGD2 in AML cells led to an obvious decrease of p-p90RSK and Bcl-2 (Fig. 6c, d). Additionally, a lower p-SHP-1 level was observed in TMIGD2 knockdown cells in comparison with control, suggesting that SHP-1 might be partially involved in TMIGD2-mediated signal transduction pathways in AML (Fig. 6d). To further confirm that CREB phosphorylation is required for the phenotypes seen in TMIGD2-depleted leukemic cells, we performed rescue experiments by introducing retrovirus encoding wild type CREB into shCtrl and shTMIGD2 HEL cells (Supplementary Fig. 6c). The ectopic expression of CREB was capable of substantially rescuing both the differentiation and colony-forming ability resulting from endogenous TMIGD2 depletion in HEL cells (Fig. 6e, f). Overall, these data suggest that the ERK1/2-p90RSK-CREB pathway functionally contributes to the effects of TMIGD2 in AML.

Since the role of TMIGD2 in AML is independent of engagement with its known ligand HHLA2 (Figs. 1d, 2d, g, h, and 7e, and Supplementary Fig. 7f), we hypothesized that TMIGD2 signaling in AML is

triggered by: (1) a new binding partner; (2) a gain-of-function mutation; (3) *trans*- and/or *cis*-homophilic dimerization. To test these hypotheses, we first performed cell-based high-throughput screening for a potential new binding partner of TMIGD2 (Supplementary Fig. 6d). Among 5477 human plasma membrane proteins and cell surface tethered human secreted proteins, as well as 371 heterodimers, no new interactor, in addition to HHLA2, was identified (Supplementary Fig. 6d). Next, to investigate the genetic factors responsible for the activation of TMIGD2 in AML, TMIGD2 cDNA of three AML specimens (Pt#19, #20 and #26) and HEL cell line were sequenced. In agreement with cBioPortal database (Supplementary Fig. 6e)[31], cDNA sequencing of cell line and primary samples showed limited alterations within intracellular domain (IC) and transmembrane domain (TM) of TMIGD2 (Supplementary Fig. 6f), suggesting that a gain-of-function mutation is unlikely the cause of TMIGD2 signaling in AML. We next determine whether TMIGD2 binds with TMIGD2 in a *trans* manner by staining HHLA2/3T3 or TMIGD2/3T3 cells with TMIGD2 mouse Fc fusion protein followed by secondary antibody staining. TMIGD2 was observed to bind with HHLA2, but not TMIGD2 (Supplementary Fig. 6g). To test whether TMIGD2 forms dimers on the same cell surface, we carried out the NanoBit proximity assay by co-transfection of constructs comprising proteins of interest fused to small-bit (SmBit) and large-bit (LgBit) (Supplementary Fig. 6h). TMIGD2-full length (FL) and TMIGD2-deleted IC (dIC), but not TMIGD2-deleted extracellular domain (dEC), generated luminescent signal, demonstrating that EC of TMIGD2 is required for *cis* homodimer formation (Fig. 6g).

To understand whether the EC and IC domains of TMIGD2 are necessary for the regulation of CREB in AML cells, TMIGD2-FL, TMIGD2-dEC, and TMIGD2-dIC were expressed in THP-1 cells followed by treatment with a CREB inhibitor 666-15[32]. THP-1 cells overexpressing TMIGD2-FL were more resistant to 666-15-induced apoptosis than cells overexpressing either TMIGD2-dEC or TMIGD2-dIC (Fig. 6h and Supplementary Fig. 6i), indicating that both EC and IC domains of TMIGD2 are important for its function in AML. We then introduced mutations at amino acids tyrosine 192 (Y192F), serine 220 (S220A), and tyrosine 222 (Y222F) within the cytoplasmic tail of TMIGD2. THP-1 cells harboring TMIGD2-Y192F and TMIGD2-S220A showed higher percentage of live cells than TMIGD2-dIC and TMIGD2-Y222F after 666-15 treatment, suggesting that tyrosine 222 is involved in TMIGD2-mediated downstream signaling pathways in AML (Supplementary Fig. 6j).

## Monoclonal antibodies against TMIGD2 demonstrate anti-leukemia activity

The development of potent and selective TMIGD2 mAbs make it possible to evaluate the therapeutic benefits of targeting TMIGD2 in AML. To this end, we generated anti-TMIGD2 mAbs recognizing distinct epitopes of TMIGD2, including clones 17C7 (mouse IgG1, mIgG1) and 20F2 (mIgG2a), which were able to disrupt the TMIGD2 *cis*-homodimerization and completely block and partially block the interaction between HHLA2 and TMIGD2, respectively (Supplementary Fig. 7a–c). When treated with 17C7 and 20F2, TMIGD2$^+$ primary AML samples showed significant decrease in colony-forming ability; however,

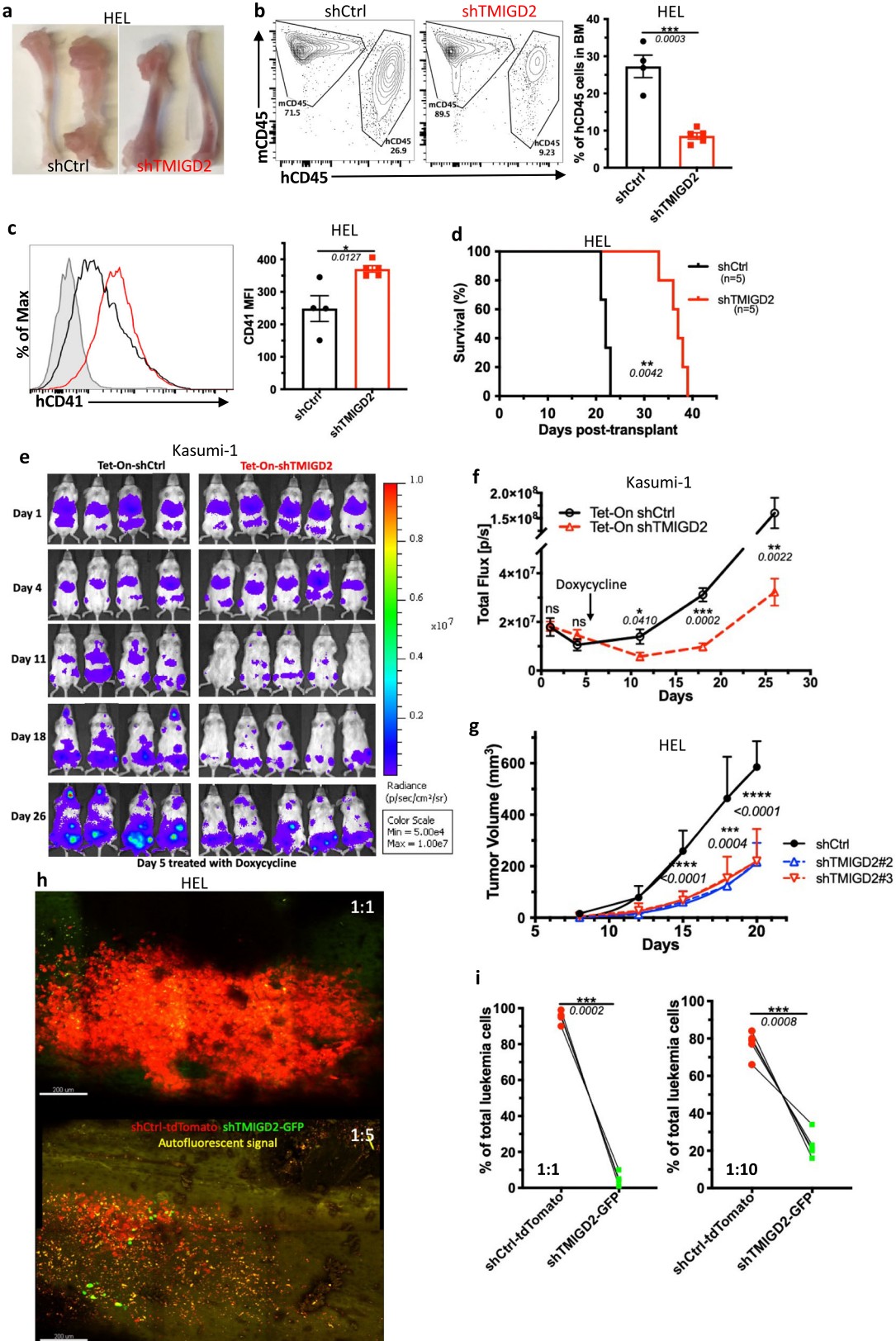

normal HSPCs from CBU and adult BM, as well as TMIGD2⁻ primary AML cells exhibited no significant difference (Fig. 7a, b and Supplementary Fig. 7d), suggesting that anti-TMIGD2 mAbs selectively inhibit TMIGD2⁺ primary AML cells. We next assessed the therapeutic efficacy of 17C7 and 20F2 in vivo using clinically relevant AML PDX models. The anti-leukemic effects of 17C7 and 20F2 were confirmed by the reduction of human CD45⁺CD33⁺ leukemia cells in peripheral blood (PB) and BM of NSG-recipients following treatments (Fig. 7c and Supplementary Fig. 7e). Notably, both 17C7 and 20F2 efficiently reduced leukemia burden in PDX model engrafted with pt#28, who had progressed after multiple previous treatments, including Bcl-2 inhibitor venetoclax (Fig. 7c).

**Fig. 3 | Loss of TMIGD2 inhibits human AML development and promotes myeloid differentiation in vivo. a** Representative images of tibia and femur from mice engrafted with shCtrl or shTMIGD2 HEL cells. **b** Representative flow cytometry plots (left) and statistics (right) of BM cells from mice inoculated with shCtrl or shTMIGD2 HEL cells. **c** Representative flow cytometry histogram (left) and quantification (right) of CD41 expression on BM hCD45$^+$ cells isolated from NSG mice engrafted with shCtrl or shTMIGD2 HEL cells. **d** Kaplan−Meier survival curves of mice transplanted with shCtrl or shTMIGD2 HEL cells ($n = 5$ per group, three independent experiments). The *p* value was calculated by the log-rank test. **e** Images of NGS mice that were engrafted with Tet-On shRNAs Kasumi-1 cells and treated with doxycycline at day 5 post engraftment to induce shRNA expression. Radiance (p/sec/cm$^2$/Sr). **f** Leukemia progression was quantified by bioluminescence. Total flux [p/s] for Tet-On shCtrl versus Tet-On shTMIGD2 Kasumi-1 cell line-derived xenograft mice. **g** Average growth curves of subcutaneous shCtrl and shTMIGD2 HEL tumors in NSG mice. **h** Representative intravital 3D flattened images of shCtrl-tdTomato and shTMIGD2-GFP HEL leukemia cells in the BM of NSG-recipients. Top, cell ratio for i.v. injection shCtrl-tdTomato: shTMIGD2-GFP = 1:1; bottom, cell ratio for i.v. injection shCtrl-tdTomato: shTMIGD2-GFP = 1:5. **i** Comparison of shCtrl-tdTomato and shTMIGD2-GFP cell percentage in the BM of NSG-recipients engrafted with mixed shCtrl-tdTomato and shTMIGD2-GFP HEL cells at the ratio of 1:1 (left) or 1:10 (right). Mean ± SEM values are shown for Fig. 3. \**p* < 0.05, \*\**p* < 0.01, \*\*\**p* < 0.001, and \*\*\*\**p* < 0.0001 by two-tailed Student's *t* test (**b**, **c**, **f**, **g**) or paired Student's *t* test (**i**). For **b**–**c** and **i**, color dots represent individual mice. Results are representative of three independent experiments. Source data are provided in the Source Data file.

By taking the advantage that both of our anti-TMIGD2 mAb 17C7 and anti-HHLA2 mAb B5B5 can completely block the interaction between HHLA2 and TMIGD2, we treated AML PDX models with either 17C7 or B5B5 to rule out the possibility that HHLA2-TMIGD2 axis contributes to the development of AML in vivo (Fig. 7d). As expected, 17C7 but not B5B5 treatment significantly reduced leukemia cells in the PB, spleen, and BM (Fig. 7e). Moreover, HHLA2 expression was undetectable in leukemia cells from BM of both mouse groups treated with 17C7 and B5B5 (Supplementary Fig. 7f). These data further confirm that the function of TMIGD2 in AML is not due to the HHLA2-TMIGD2 interaction.

As 20F2 is a mouse IgG2a mAb, we next explored whether its anti-leukemia effects could be partially attributed to antibody-dependent cellular phagocytosis (ADCP) by performing a BM-derived macrophage coculture system. 20F2 opsonized primary leukemia cells resulted in a significantly higher ADCP activity compared with control mIgG2a (Fig. 7f), suggesting that 20F2 may both block oncogenic TMIGD2 signaling and induce ADCP. To further analyze whether the direct involvement of macrophages that elicit ADCP in vivo is necessary for 20F2 mAb therapeutic efficacy, we treated PDX NSG mice with anti-CSF1R mAb to deplete macrophages (Fig. 7g and Supplementary Fig. 7g). Interestingly, the depletion of macrophages had limited effect on 20F2 therapy (Fig. 7h), indicating that macrophages and ADCP are not the major anti-leukemia effectors mediated by 20F2 in vivo. Since 20F2 could selectively block the HHLA2-independent TMIGD2 signaling in AML cells but largely maintain costimulatory signaling mediated by HHLA2-TMIGD2 interaction on T and NK cells (Supplementary Fig. 7h, i), which is potentially better than 17C7, we treated more AML PDX models that were established with hard-to-treat or secondary AML patient samples (Pt#19, Pt#8 and Pt#24), and found that all responded well to 20F2 therapy (Fig. 7i). Next, we tested 20F2 in the first round PDX models established with AML patient samples representing three subsets of AML based on TMIGD2 expression, including CD34$^+$TMIGD2$^{low}$ (Pt#31, <10% TMIGD2$^+$ cells), CD34$^+$TMIGD2$^{high}$ (Pt#28, >80% TMIGD2$^+$ cells) and CD34$^-$TMIGD2$^+$ (Pt#21, 37% TMIGD2$^+$ cells). The first round PDX mice were treated with isotype control versus 20F2. Downregulation of TMIGD2 and lower percentage of CD34$^+$CD38$^-$ cells were detected in 20F2 treated group compared with isotype control (Supplementary Fig. 7j,k). The second round NSG-recipient were xenotransplanted intravenously with FACS-purified CD33$^+$ myeloid cells isolated from the first round AML PDX mice. The second round NSG-recipient received no further 20F2 treatment. We observed significant reduced leukemia burden in both rounds of PDX mice (Fig. 7j), indicating that anti-TMIGD2 mAb preferentially inhibit leukemogenesis by targeting TMIGD2-expressing LSCs.

To test whether the sensitivity of normal HSPCs to 20F2 was the same as AML samples, 20F2 was used to treat NSG-recipients engrafted with CD34$^+$ cells from cord blood. In contrast to the observation in AML PDX models, 20F2 showed negligible inhibitory effects on the engraftment of normal cells, including total hCD45$^+$ cells, hCD19$^+$ lymphoid cells, and CD33$^+$ myeloid cells (Fig. 7k, Supplementary Fig. 7l), which could be a consequence of relatively low surface TMIGD2 expression and intrinsic relative TMIGD2-signaling independence in HSPCs. Taken together, targeting TMIGD2 with mAb differentially eliminates AML cells over normal HSPCs, and is a promising therapeutic strategy.

## Discussion

LSCs are considered as the root cause of the initiation and progression of AML, as well as treatment resistance and relapse of the disease[5,6,33]. Thus, identifying targetable molecules that are differentially expressed on LSCs and their normal counterparts is of great importance. Previously, various surface proteins have been identified as LSC-specific markers, such as CD47[34,35], CD70[36,37], CD93[38], CD99[39], CD123[24], IL1RAP[23,40], and Tim-3[41]. However, there was limited data to support their roles as cell-intrinsic regulators of LSCs, with Tim-3 supporting LSCs by a Galectin-9/Tim-3 autocrine loop[2,6,42,43]. Moreover, it is important to note that LSC frequency and phenotypic diversity dramatically change after treatment and progression, indicating that targeting a single molecule is insufficient to completely eliminate LSCs[7]. In the present study, we identified TMIGD2 as a cell surface marker aberrantly overexpressed on LSCs in comparison with normal HSPCs and hypothesized that TMIGD2 is more than a passive LSC marker, but rather is required for LSC maintenance and leukemogenesis. Consistent with this hypothesis, we showed that higher TMIGD2 expression in AML was correlated with worse overall survival. Transcription factor RFX1 is potentially involved in the triggering of TMIGD2 overexpression in LSCs: first, RFX1 can bind the open chromatin region of TMIGD2[44]; second, TMIGD2 expression is positively correlated with RFX1 expression at mRNA level[20]; third, RFX1 has been shown to be enriched in FLT3-ITD-specific subclones derived from AML patients, which are able to expand in vitro and engraft in vivo with higher proliferative capacity than the FLT3-WT subclones[11]. We also demonstrated that knockdown of TMIGD2 using shRNA or targeting TMIGD2 with mAbs significantly decreased colony-forming capacity of LSCs and effectively reduced leukemia burden in PDX models, highlighting TMIGD2 as a novel therapeutic target for AML treatment.

In efforts to accurately define LSCs and predict initial therapy resistance, several LSC signatures including LSC17 and LSC-related (LSC-R) gene profile, have been established by xenotransplantation[12,45]. Importantly, our RNA-seq analysis on paired patient samples (CD34$^+$TMIGD2$^+$ and CD34$^+$TMIGD2$^-$) clearly demonstrated that genes involved in LSC17 and LSC-R signatures were highly enriched in CD34$^+$TMIGD2$^+$ cells, indicating that TMIGD2 may be served as a prognostic marker for quantification of minimal residual disease. Our observation that higher frequency of LSCs resided in TMIGD2$^+$ subpopulation as compared to TMIGD2$^-$ counterpart led us to consider that invoking TMIGD2 in AML might be required for LSC maintenance. Indeed, we found that knockdown of TMIGD2 in primary AML cells impaired the stem and progenitor

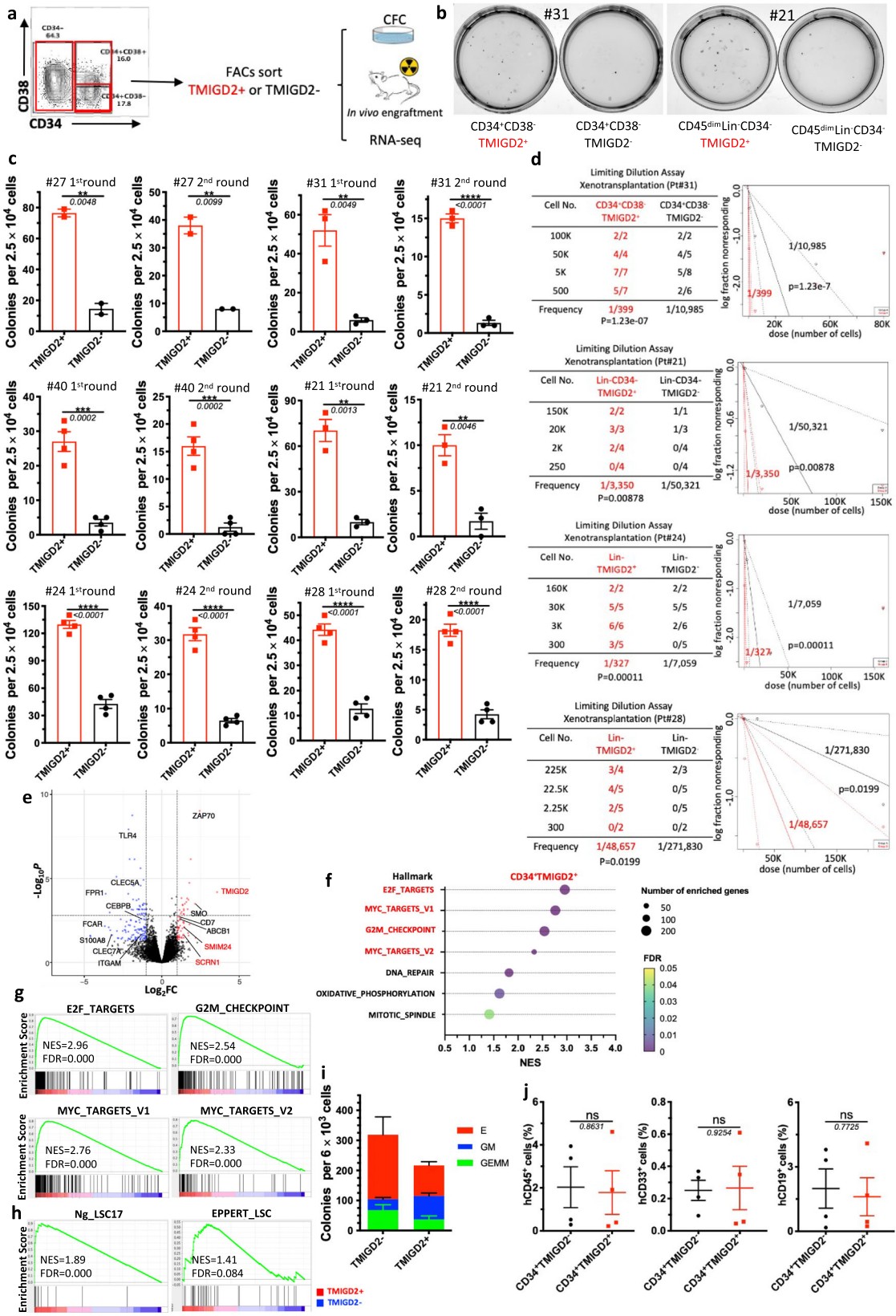

potential of AML and inhibited their development in NSG mice. In contrast to its role in LSCs, TMIGD2 appeared to play a minor role in normal HSPCs. First, limited TMIGD2 was expressed on HSPCs. Second, targeting TMIGD2 with mAbs or knockdown of TMIGD2 exhibited slight effect on normal HSPCs self-renewal in vitro and hematopoiesis in vivo, suggesting that TMIGD2 is a feasible target for AML therapy.

TMIGD2 was previously identified as a co-stimulatory receptor as well as an adhesion molecule[14–16]. HHLA2 is the known ligand for TMIGD2 expressed on T and NK cells. However, we found that

**Fig. 4 | TMIGD2 enriches for functional LSCs. a** Schematic for flow cytometry sorting and subsequent CFC assay and in vivo xenotransplantation. The highest (TMIGD2⁺) and lowest (TMIGD2⁻) TMIGD2-expressing blasts (top and bottom 15%, respectively) from the CD34⁺CD38⁻, CD34⁺CD38⁺ and CD34⁻ AML specimens were FACS-purified and used for CFC assays, xenotransplantation and RNA-seq analysis. **b** Representative photo of CFC plates seeded with FACS-purified TMIGD2⁺ and corresponding TMIGD2⁻ AML patient cells with respect to colony formation after 12 days culture. Left, Pt#31; right, Pt#21. **c** First round (left) and second round (right) CFC assays using FACS-purified TMIGD2⁺ or TMIGD2⁻ cells of CD34⁺CD38⁻ (Pt#27 and Pt#31), CD34⁺CD38⁺ (Pt#40), CD34⁻ (Pt#21) and Lin⁻ (Pt#24 and Pt#28) sub-populations from individual AML patients. **d** Limiting dilution assay. Table (left) showing different numbers of FACS-purified cells and recipient NSG mice used for the xenotransplantation. Graph (right) showing the frequency of LSC in TMIGD2⁺ and TMIGD2⁻ subpopulations. The limiting dilution analysis was performed using ELDA (Extreme Limiting Dilution Analysis) software. **e** The volcano plot showing differentially expressed genes from RNA-seq analysis on CD34⁺TMIGD2⁺ and CD34⁺TMIGD2⁻ cells sorted from six primary AML patient samples. **f** Scattergrams of the top pathways that were significantly enriched in CD34⁺TMIGD2⁺ cells based on GSEA. **g** GSEA plots showing enrichment of gene sets for E2F targets, Myc targets and G2M checkpoints in CD34⁺TMIGD2⁺ versus CD34⁺TMIGD2⁻ groups from six primary AML patient samples. **h** GSEA plots showing the stemness signatures were enriched in CD34⁺TMIGD2⁺ cells. **i** CFC assay was performed with FACS-purified TMIGD2⁺ or TMIGD2⁻ cells from normal CD34⁺ HSPCs. **j** Percentages of hCD45⁺, hCD33⁺ and hCD19⁺ cells in the BM of NSG-recipient mice at 3 months post trans-plantation of FACS-purified TMIGD2⁺ or TMIGD2⁻ cells from CD34⁺ HSPCs. Mean ± SEM values are shown for Fig. 4c, i, j. ns by two-tailed Student's *t* test. Normalized enrichment score (NES) and false discovery rate (FDR) are shown for Fig. 4g, h. Color dots in **c** and **i** represent technical replicates. Color dots in **j** represent individual mice. Results are representative of three independent experiments. Source data are provided in the Source Data file.

HHLA2 was not needed for TMIGD2 expressed on LSCs. Further-more, no new ligand for TMIGD2 was identified by performing a human cell microarray screen, including 5477 human plasma membrane proteins and cell surface tethered human secreted proteins as well as 371 heterodimers, in an attempt to explain the phenotype with potential ligand-receptor interaction. TMIGD2 was previously reported to regulate endothelial cell-cell adhesion and barrier as well as angiogenesis by forming *trans-* and *cis*-homophilic dimers, resulting in phosphorylation of Ser220[46]. However, we were unable to detect TMIGD2-TMIGD2 binding in *trans* between two cells. We demonstrated that TMIGD2 bound with TMIGD2 on the same cell surface in a *cis* interaction. Notably, both extracellular and intracellular domains of TMIGD2 were essential for its function in AML. Our study provided new insight into the role and mechanism of TMIGD2 in AML.

We further dissected the mechanism underlying the dependency of LSCs on the function of TMIGD2. By performing a phospho-kinase array and western blots, CREB was identified as a key transcription factor downstream of the TMIGD2 signaling pathway in AML cells. Previous studies have shown that CREB overexpression leads to upregulation of genes that control cell proliferation, cell-cycle and differentiation, and that high levels of CREB is associated with an increased risk of relapse and poor clinical outcome[29]. Consistent with our results, CREB can be activated through phosphorylation by p90RSK, and upregulates *cyclins* and *Bcl2*[30]. Remarkably, our rescue assay by overexpressing CREB in TMIGD2 knockdown cells largely restored the effects of TMIGD2 knockdown on cell self-renewal and differentiation, confirming that CREB was a bona fide functionally important target of TMIGD2 in AML. Of note, other signal transducers might also partially mediate the functions of TMIGD2 in AML, such as SHP-1, as evidenced by dramatically decreased level of p-SHP-1 upon TMIGD2 knockdown. Recent studies have shown that genetic deple-tion or chemical inhibition of SHP-1 enhances AML differentiation and compromise leukemogenesis through STAT6-*Itgb3* (encode CD61) axis[47,48]. Our finding that TMIGD2 knockdown increased the expression of differentiation markers CD61 and CD41, provided a mechanistic explanation for the observed phenotype.

Finally, among a panel of anti-TMIGD2 mAbs generated by us, clones 17C7 and 20F2 demonstrated therapeutically effects in vitro and in vivo by using primary human AML samples. Meanwhile, the treatment showed negligible impact on normal HSPCs that had limited TMIGD2 expression and might largely maintain the co-stimulatory signaling mediated by TMIGD2 on NK and T cells. Overall, our studies have elucidated novel roles and mechanisms of TMIGD2 in LSCs. TMIGD2 knockdown and anti-TMIGD2 mAb treatment compromised LSCs and impeded leukemogenesis in PDX models of refractory/relapsed AML, underscoring the therapeutic value of targeting TMIGD2 in AML.

## Methods

### Leukemic patient samples and normal hematopoietic cell samples

Human AML samples and healthy donor samples were obtained with informed consent at Montefiore Medical Center/Albert Einstein Cancer Center in congruence with the protocol approved by the institutional review board (IRB# 11-02-060E). Characteristics of AML patients were outlined in Supplementary Table 1. Human cord blood units were purchased from the New York Blood Center. Leukemia blasts and mononuclear cells (MNCs) were isolated using Lymphoprep (STEMCELL Technologies) density gradient separation and CD34⁺ cells were enriched using the CD34 Microbead kit (130-046-702, Miltenyi Biotec).

### Cell culture

For human leukemic cell lines, Kasumi-1, HEL, K562, Kg-1a, ME-1, and THP-1 were obtained as gifts from Dr. Ulrich Steidl lab at Albert Einstein College of Medicine and cultured in RPMI supplemented with 10% FBS and 1% penicillin & streptomycin. Human cord blood CD34⁺ cells and human primary leukemic cells were maintained in StemSpan SFEM II (STEMCELL Technologies) supplemented with primocin (100 μg/ml, InvivoGen), recombinant human TPO (100 ng/ml), recombinant human FLT3-L (100 ng/ml), recombinant human SCF (100 ng/ml), recombinant human IL-3 (20 ng/ml), and recombinant human IL-6 (20 ng/ml) (BioLegend).

### Mice

NOD/SCID IL2Rgamma^null (NSG) and BALB/c female mice, 6-8 weeks old, were purchased from Jackson Laboratory and Charles River, respectively. The animals were housed in a specific pathogen free facility, in 12 hour light/12 h dark cycles with temperatures maintained between 65 F to 75 F, and humidity maintained 40−60%. Littermates of the same sex were randomized to experimental groups. Mice were bred and maintained in individual ventilated cages and fed with autoclaved food and water at the animal facility at Albert Einstein College of Medicine. All mouse studies were performed in compliance with approved protocols from the Institutional Animal Care and Use Committee at Albert Einstein College of Medicine.

### Flow cytometry and cell sorting

Primary leukemic cells and normal MNCs were stained with anti-human CD45 to identify blasts, anti-human CD19 and anti-human CD3 to exclude lymphocytes, and other surface markers, including CD14, CD33, CD34, CD38, CD45RA, CD123, IL1RAP, HHLA2, and TMIGD2 to define distinct subpopulations. Cell viability was assessed with DAPI. Intracellular staining was performed to stain p-ERK1/2 and p-CREB in NBM, CBU and primary AML cells. Flow cytometry data was collected on a BD LSRII flow cytometer and analyzed using FlowJo software (BD Biosciences). Cell sorting was performed on a BD FACS Aria II.

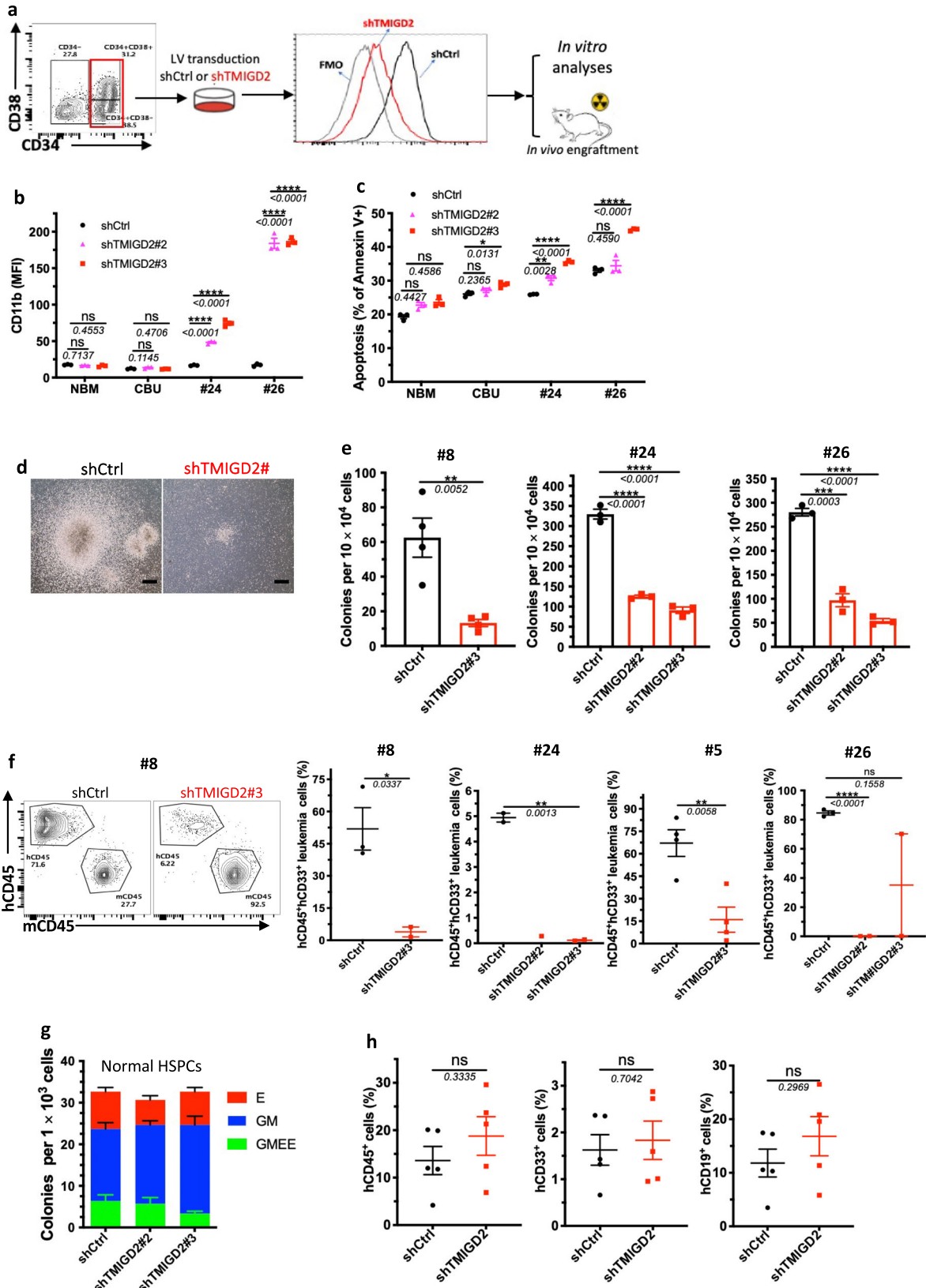

## RT-qPCR

Total RNA was isolated from cell pellets using RNeasy Plus Mini Kit (QIAGEN) following the manufacturer's instructions and reverse-transcribed into cDNA with SuperScript IV First-Strand Synthesis System (Thermo Fisher Scientific). Quantitative real-time PCR analysis was performed with TaqMan Gene Expression Master Mix (Thermo Fisher Scientific) in a 7900 HT System (Applied Bio-system). TaqMan primers targeting *HHLA2* (Hs00737670_m1), *TMIGD2* (Hs00758270_m1), *GAPDH* (Hs03929097_g1), and *GUSB* (Hs00939627_m1) were purchased from Thermo Fisher Scientific.

**Fig. 5 | Loss of TMIGD2 impairs LSC maintenance and prevents human AML expansion in vivo. a** Schematic for flow cytometry sorting strategy, lentivirus transduction and subsequent experimental design of in vitro and in vivo assays. **b** Expression of myeloid lineage marker CD11b after knockdown of TMIGD2 in primary CD34+ cells isolated from NBM, CBU and AML patient samples (Pt#24 and Pt#26). **c** Percentage of Annexin V+ cells in shTMIGD2 compared with shCtrl primary CD34+ cells as assessed by flow cytometry. **d, e** Representative morphology (**d**, Pt#8) and quantification (**e**) of colonies derived from sorted primary CD34+ AML cells transduced with lentiviruses targeting shCtrl or shTMIGD2. Scale bar: 100 μm for colonies. **f** Representative flow cytometry plots (left) and engraftment potential (right) of PDX mouse models established with primary AML cells that were transduced with lentiviruses targeting shCtrl or shTMIGD2. **g** CFC assay using normal CD34+ HSPCs upon knockdown of TMIGD2 (n = 3). **h** Percentages of hCD45+, hCD33+ and hCD19+ cells in the BM of NSG-recipient mice at 3 months post transplantation of shCtrl or shTMIGD2 CD34+ HSPCs. Mean ± SEM values are shown for Fig. 5. ns, \*p < 0.05, \*\*p < 0.01, \*\*\*p < 0.001, and \*\*\*\*p < 0.0001 by two-tailed Student's t test. Color dots in **b, c, e, g** represent technical replicates. Color dots in **f** and **h** represent individual mice. Results are representative of three independent experiments. Source data are provided in the Source Data file.

Target gene expression were quantitated utilizing the comparative CT (ddCT) method and normalized against the housekeeping genes *GAPDH* and *GUSB*.

## Lentiviral-mediated shRNA knockdown

Stable knockdown in human AML cell lines was achieved by using the pLKO.1-shRNAs targeting human TMIGD2 (shTMIGD2#1 5′-CCGGCAT CCTGTGTCAACCGTACATCTCGAGATGTACGGTTGACACAGGATGTT TTTG, shTMIGD2#2 5′-CCGGGCATTCTACAGCAACGTCCTACTCGAGT AGGACGTTGCTG TAGAATGCTTTTTG-3′ and shTMIGD2#3 5′-CCGG GCTCCGTGTTAAGTGGACAAACTCGAGTTTGTCCACTTAACACGGAG CTTTTTG-3′), HHLA2 (shHHLA2#1 5′-CCGGCCTTTGGCTTTCTTCATTT ATCTCGAGATAAATGAAGAAAGCCAAAGGTTTTTG-3′, shHHLA2#2 5′-CCGGGAAGACTTGATGAAGATATAACTCGAGTTATATCTTCATCAAGT CTTCTTTTTG-3′, and shHHLA2#3 5′-CCGGACACAAACAGCTTCTTA ATATCTCGAGATATTAAGAAGCTGTTTGTGTTTTTTG-3′), and a non-targeting control shRNA (shCtrl 5′-CCGGCCTAAGGTTAAGTCGCCCTC GCTCGAGCGAGGGCGACTTAACCTT AGGTTTTTG-3′). HEK 293 T cells were transfected with 6 μg of lentiviral constructs, 3.6 μg of packaging plasmid psPAX2 (Plasmid #12260) and 2.4 μg of pCMV-VSV-G (Plasmid #8454) with JetPRIME (Polyplus) transfection reagent. For knockdown of TMIGD2 in primary cells, pSIH1-H1-copGFP shRNA vector (System Biosciences) was used to produce lentivirus following the manufacturer's instructions. Lentiviral supernatants were harvested and concentrated at 48 and 96 hours using Lenti-X concentrator (Clontech). Human primary cells and cell lines were transduced by spin infection with concentrated lentivirus. Transduced AML cell lines were treated with 2.5 μg puromycin (InvivoGen) for 3 days. Primary cells expressing GFP were flow sorted 30 h after spin infection. Knockdown efficiency was evaluated by flow cytometry.

## Cell cycle and apoptosis assays

DAPI and anti-human ki-67 antibody were used to assess the cells located at G0, G1, and S/G2/M stages. For apoptosis assays, single-cell suspensions in Annexin V binding buffer (BioLgend) were stained with Annexin V (BioLgend) and DAPI (BioLgend). Data was collected using a BD LSRII flow cytometer.

## Colony formation and serial replating assays

AML cell lines were seeded at a density of 300-1000 cells per 35 mm dish in 1.2 ml methylcellulose serum-free base media (HSC002SF, R&D). Colonies was counted after 10-12 days of incubation at 37 °C. For the serial replating assay, colonies were collected and 500 cells were subsequently replated in fresh methylcellulose media. For primary cells, CFC assays were performed to investigate the frequency, self-renewal, and differentiation of hematopoietic and leukemic stem cells in semisolid media. 1,000-30,000 transduced or flow-sorted primary CD34+ cells were mixed in 1.2 ml methylcellulose enriched media (HSC005, R&D) and incubated for 12-14 days, followed by enumeration of the size, morphology, and number. Then, colony cells were collected and replated every 12 days. For the CFC assays of primary CD34+ cells treated with mAbs, 1,000-30,000 cells were seeded in 1.2 ml methylcellulose enriched media,

supplemented with 50-100 μg of anti-TMIGD2 mAbs or isotype controls.

## Cell line- and primary human AML patient-derived xenograft mouse model

To generate AML cell line-derived xenograft (CDX) mouse model, 200K-500K transduced shCtrl or shTMIGD2 cell lines were injected intravenously into sublethally irradiated (200 cGy) 6-8 weeks old NSG mice via tail vein. Mice were followed for up to 6-week post injection for chimerism analysis and survival. BM and spleen cells were analyzed by flow cytometry. One million Tet-On Kasumi-1-luciferase cells were intravenously injected into NSG mice and the leukemia progression was measured weekly by the in vivo imaging system in Einstein. For subcutaneous xenograft tumor model, one million of shCtrl or shTMIGD2 AML cell lines were implanted subcutaneously in the flank of NSG mice. Tumor volume was calculated using the standard tumor formula V = (length * width²)/2. For all leukemia xenografts, subcutaneous tumors were allowed to grow until the humane endpoints (volume 2000 mm³) or mice showing sick or moribund status, upon which the animals were sacrificed. NSG mice intravenously injected with leukemia cells were sacrificed when clinical signs of leukemia were observed (hind limb paralysis), or they were moribund.

For anti-TMIGD2 mAbs treatment of newly-established AML PDX models, sublethally irradiated (200 cGy) female NSG mice were transplanted with CD34+ primary AML cells followed by in vivo treatment with isotype control versus anti-TMIGD2 mAbs (200 μg, i.p. every 3 days). PB, spleen, and BM cells were collected for flow cytometry analysis post-transplantation at time-points as indicated in the figure legends. The second round PDX mice were intravenously injected with FACS-purified human CD33+ cells isolated from the BM of primary PDX mice. The second round NSG-recipient received no further anti-TMIGD2 mAb treatment. Pt#31, primary engraftment 500k cells, secondary engraftment 500k cells; pt#28, primary engraftment 500k cells, secondary engraftment 50k cells; pt#21, primary engraftment 500k cells, secondary engraftment 60k cells.

## Intravital imaging and analysis

Mice were anesthetized using isoflurane and secured on a warming plate. The tibial bone was surgically exposed and thinned as previously described[49]. All imaging was performed on an Olympus FVE-1200 upright microscope, using 25×1.04 NA objective, and Deepsee MaiTai Ti-Sapphire pulsed laser (Spectra-Physics) tuned to 920 nm. Mice were imaged in a custom-built 37 °C-heated incubator chamber to maintain body temperature. Time-lapse movies were conducted every 1 min for at least 1 hour with 3 μm z spacing. Qtracker 705 (Invitrogen) was used to highlight vascular region and injected at the start of imaging.

All image analysis was conducted using Imaris 9.3 (Bitplane) to track cells and correct drift. For drift correction, autofluorescent sessile macrophages were tracked using semi-automated or manual methods to correct XYZ registrations over time. Ratio channels were

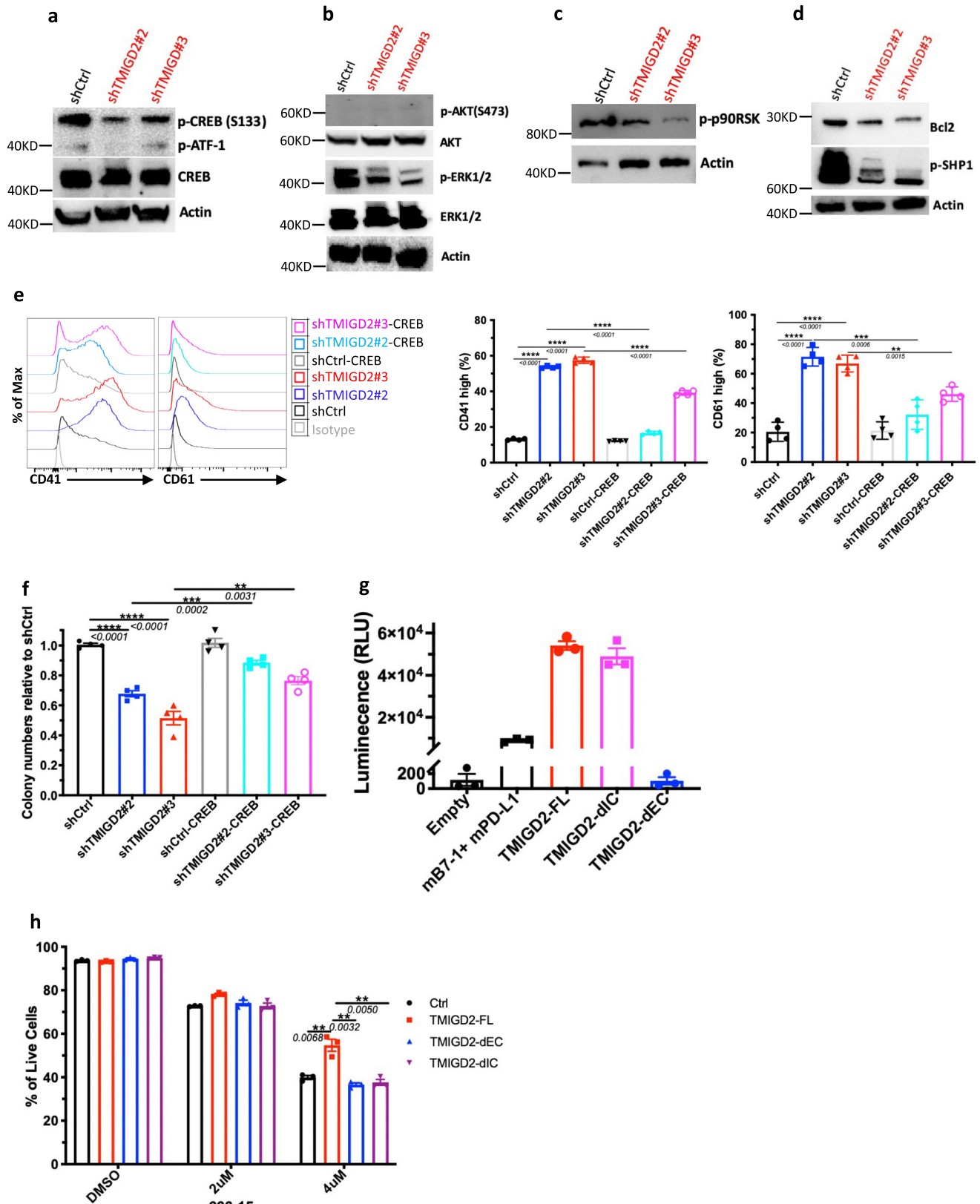

used to isolate shCtrl-tdTomato and shTMIGD2-GFP AML cells and subtracted channels were created to isolate vessel signal. Maximal intensity time projection was used to highlight the AML traces over time for the overlap with vascular region. All imaging and conditions were independently conducted at least twice.

**Limiting dilution assays**

For in vivo limiting dilution assay, the flow-sorted TMIGD2+ and TMIGD2- primary AML cells were injected intravenously into sub-lethally irradiated (200 cGy) 6-8 weeks old female NSG-recipient mice via tail vein with four different doses of cells for each group.

**Fig. 6 | TMIGD2 activates multiple signal transduction pathways in AML through the transcription factor CREB. a** Western blot analysis of phospho-CREB (p-CREB) and total CREB level changes in HEL cells upon TMIGD2 knockdown. **b**–**d** Western blot images showing the level changes of p-ERK1/2 (**b**), p-AKT (**b**), p-p90RSK (**c**), Bcl2 (**d**), and p-SHP1 (**d**) in shCtrl versus shTMIGD2 HEL cells. **e** Representative flow cytometry histograms (left) and quantification (right) of CD41 and CD61 expression on HEL cells engineered to express CREB with or without TMIGD2 knockdown. **f** Colony-forming ability of HEL cells from Fig. 6e (500 cells/plate). **g** NanoBit proximity assay showing TMIGD2 binds with TMIGD2

on the same cell surface in a *cis* interaction. Mouse PD-L1 (mPD-L1) and mB7-1 form a heterodimer, which was served as a positive control. TMIGD2-FL, full length TMIGD2. TMIGD2-dEC, a mutant TMIGD2 with deleted extracellular domain. TMIGD2-dIC, a mutant TMIGD2 with deleted intracellular domain. **h** Effect of a CREB inhibitor 666-15 on apoptosis of THP-1 cells overexpressing vector Ctrl, TMIGD2-FL, TMIGD2-dEC and TMIGD2-dIC. Mean ± SEM values are shown for Fig. 6. **p < 0.01, ***p < 0.001, and ****p < 0.0001 by two-tailed Student's *t* test. Color dots in **e**–**h** represent technical replicates. Results are representative of three independent experiments. Source data are provided in the Source Data file.

Leukemic engraftment was defined as >1% detectable human cells in the BM 12 weeks post xenotransplantation. Leukemia initiating cell frequency was calculated using the ELDA method (https://bioinf.wehi.edu.au/software/elda/).

## RNA sequencing and data analysis

For RNA-seq using primary AML cells, total RNA was extracted from the flow-sorted CD34+TMIGD2+ and CD34+TMIGD2- cells using RNeasy Plus Micro Kit (QIAGEN). cDNA libraries were generated using the SMARTer Ultra Low RNA kit (Clontech) for Illumina sequencing with ≥40 million paired-end reads per sample by Admera Health. Likewise, total RNA was isolated from the shCtrl and shTMIGD2 HEL cells with RNeasy Plus Mini Kit (QIAGEN) following the manufacturer's instructions. RNA-seq libraries were prepared using the NEBNext Ultra II directional RNA workflow with Poly (A) mRNA enrichment for Illumina sequencing by Admera Health. RNA-seq reads from both the TMIGD2 knockdown HEL cells and the flow-sorted primary AML cells were aligned to the human genome (hg38) using the program STAR (version 2.6.1b)[50]. HTSeq (version 0.6.1) was then used to count the number of RNA-seq fragments in each gene in the refSeq annotation (downloaded from the UCSC genome browser in 11/2019)[51]. Next, DESeq2 (version 3.11) was utilized to perform differential expression analysis and principal component analysis (PCA) on genes that had an average read count ≥1 in either of the two compared sample groups[52]. To identify enriched gene sets, GSEA was performed on the ranked gene list[53]; specifically, ranked genes were calculated by $\log_{10}$ (p-value) multiplied by the sign of the $\log_2$ (fold change).

## Phospho-kinase array and western blot

The human phosphor-kinase array (ARY003B, R&D Systems) was performed following the manufacture's protocol. Specifically, shCtrl and shTMIGD2 HEL cell lysates were incubated with the array membranes overnight at 4 °C on a rocking platform shaker. The membranes were then incubated with biotinylated antibody cocktails, followed by horseradish peroxidase (HRP) conjugated streptavidin. The membranes were exposed with multiple exposure times using the ChemiDoc Imaging System (Bio-Rad) after incubation with chemiluminescent detection reagent mix.

Western blots analysis was performed by means of standard techniques. Briefly, cells were washed with ice-cold PBS and lysed in RIPA lysis buffer supplemented with phosphatase and protease inhibitor cocktails (Thermo Fisher Scientific). Cell extracts were centrifuged and the supernatants were collected and quantified by the BCA method (Thermo Fisher Scientific). Equal amounts of lysates were loaded and separated by 12% SDS-PAGE and transferred onto PVDF membranes. The membranes were blocked and incubated sequentially with desired concentrations of primary and secondary antibodies, followed by detection with the Pierce ECL substrate.

## Rescue assay

MSCV-CREB-YFP and MSCV-YFP constructs were generated using Gibson Assembly Kit (NEB) following the manufacture's protocol. Retrovirus were produced by co-transfecting Phoenix cells with pCMV-

VSV-G (Plasmid #8454) and MSCV vectors using JetPRIME transfection reagent. HEL cells were spin infected with retrovirus supplemented with 8 µg polybrene (Sigma-Aldrich). YFP+ cells were then flow-sorted 30 hours post transduction. Total CREB and phospho-CREB levels were validated by western blot.

## Generation of mAbs against TMIGD2 and HHLA2

Mouse anti-human TMIGD2 mAbs and mouse anti-HHLA2 mAbs were generated by standard hybridoma techniques[13,17]. Splenocytes from TMIGD2-Ig or HHLA2-Ig immunized BALB/c mice were fused to Bcl-2-overexpressing Ag8 myeloma cells. High-throughput flow cytometry (BD Canto II) was then performed to screen clones that specifically recognized HHLA2 and TMIGD2.

## Human receptor interaction microarray screening

Screening for potential ligands of TMIGD2 was performed using the Retrogenix Cell Microarray technology. 10 µg/ml of human TMIGD2-hFc was screened for binding against fixed HEK293 cells/slides individually over-expressing duplicate 5477 human plasma membrane proteins and cell surface tethered human secreted proteins, as well as 371 heterodimers (16 slide sites, n = 2 slides per slide set). All transfection efficiencies exceeded the minimum threshold. Detection of binding was performed using an AlexaFluor 647 anti-hIgG Fc secondary antibody. In total, 9 primary hits were identified by analyzing fluorescence on ImageQuant software (GE). A confirmation/specificity screen was done to determine which hits were specific for TMIGD2.

## Plate-based protein binding assays

Nunc Maxisorp ELISA Plates (BioLegend) were coated with recombinant proteins Galectin-1 and Galectin-9 at 2 µg/ml in binding buffer at 4 °C overnight. Wells were then washed and blocked with 2% BSA in PBS, followed by incubation with serial dilutions of hTim-3-hFc, TMIGD2-hFc, and hPD-L1-hFc fusion proteins (R&D). Next, HRP-labeled goat anti-human IgG (Southern Biotech) was incubated with plates after washes. TMB substrate solution (Thermo Fisher Scientific) was added and the reaction was finally terminated by the addition of $H_2SO_4$. Absorbances at 450 nm with a reference wavelength of 570 nm was then determined using a Wallac 1420 Victor2 Microplate Reader.

## NanoBit proximity assay

HEK 293 T cells were transfected with constructs (Promega) of interest with JetPRIME transfection reagent. Two days post-transfection, Nano-Glo live cell substrate (Promega) was added following the manufacture's protocol. Luminescence signal was collected using Wallac 1420 Victor2 Microplate Reader.

## In vitro ADCP assay

BM-derived macrophages (BMDMs) were isolated from tibia and femur of NSG mouse, and then cultured in DMEM medium supplemented with 50 ng/ml mouse M-CSF (Biolgend) for 7 days. Primary leukemia cells were pre-labeled by CFSE (Thermo Fisher) according to the manufacture's protocol, and incubated with indicated concentration of 20F2 or isotype control for 30 minutes at 37 °C. BMDMs were then added to primary leukemia cells for coculture at a 3:1 ratio of leukemia

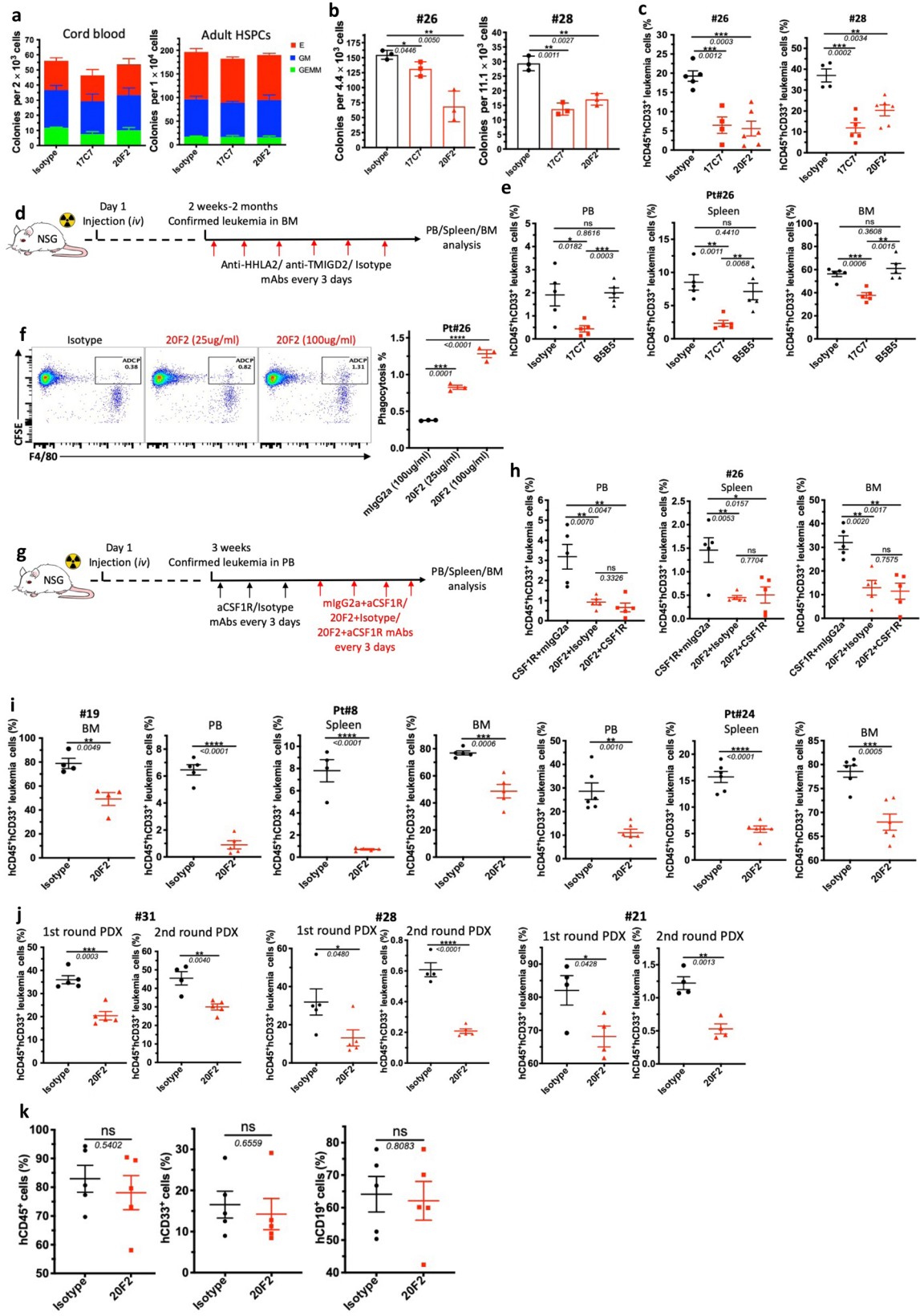

cells/BMDMs for 20 hours at 37 °C. Subsequently, cells were analyzed using flow cytometry.

## Statistics and reproducibility

Statistical analyses were performed using GraphPad Prism 9 (GraphPad Software, Inc.) and were presented as mean ± SEM or mean ± SD as indicated. Two-tailed Student's *t* test or paired Student's *t* test was used to compare means between groups as indicated; $p < 0.05$ was considered statistically significant. Kaplan-Meier survival curves were plotted with GraphPad Prism 9 and the p values were calculated using the log-rank (Mantel Cox) test. For immunoblots and treatment assays, the experiments have been

**Fig. 7 | Anti-TMIGD2 mAbs reduces leukemia burden in AML PDX mouse models but does not affect normal human HSPCs in NSG mice. a** CFC assay using CBU (left) and NBM (right) -derived CD34$^+$ HSPCs treated with isotype control or anti-TMIGD2 mAbs (17C7 and 20F2). **b** CFC counts of primary CD34$^+$ AML cells after treatment with isotype control or anti-TMIGD2 mAbs (17C7 and 20F2). **c** Percentage of hCD45$^+$CD33$^+$ leukemia cells in PB (Pt#26) and BM (Pt#28) of AML PDX mouse models after treatment with isotype control or anti-TMIGD2 mAbs (17C7 and 20F2). **d** Schematic for engraftment of primary AML specimens and in vivo treatment with anti-HHLA2, anti-TMIGD2 and isotype mAbs. **e** Frequency of hCD45$^+$CD33$^+$ leukemia cells in PB, spleen and BM after treatment with mIgG1 isotype control, anti-HHLA2 (B5B5, mIgG1) and anti-TMIGD2 (17C7, mIgG1) mAbs. **f** Phagocytosis of primary AML cells by mouse BM derived macrophages in the presence of 20F2 or mIgG2a isotype control. **g** Experimental schematic of in vivo macrophage depletion with anti-CSF1R treatment (black arrow), followed by 20F2 and anti-CSF1R mAbs

combination treatment (red arrow). aCSF1R, anti-CSF1R. **h** Percentage of hCD45$^+$CD33$^+$ leukemia cells in PB, spleen and BM in Fig. 7g. **i** Percentage of hCD45$^+$CD33$^+$ leukemia cells in PB, spleen and BM of AML PDX mouse models established with R/R AML samples after treatment with isotype control or 20F2. **j** Frequency of hCD45$^+$CD33$^+$ leukemia cells in the BM of the first round (left) and second round (right) PDX mouse models. The primary PDX mice were treated with isotype control versus 20F2. The second round NSG-recipients received no further 20F2 treatment. **k** Percentage of human CD45$^+$, CD33$^+$ and CD19$^+$ cells in the BM of NSG-recipients engrafted with normal HSPCs isolated form cord blood after treatment with mIgG2a isotype control or 20F2. Mean ± SEM values are shown for Fig. 7. ns, *$p < 0.05$, **$p < 0.01$, ***$p < 0.001$, and ****$p < 0.0001$ by two-tailed Student's $t$ test. Color dots in **a**, **b**, and **f** represent technical replicates. Color dots in **c**, **e**, **h** and **i**–**k** represent individual mice. Results are representative of three independent experiments. Source data are provided in the Source Data file.

repeated at least three with similar results and representative data was shown.

### Reporting summary
Further information on research design is available in the Nature Portfolio Reporting Summary linked to this article.

### Data availability
All unique reagents generated in this study are available upon request. All requests need to execute a suitable Materials Transfer Agreement. The RNA-seq data generated in Figs. 2 and 4 have been deposited in the Gene Expression Omnibus (GEO), with accession numbers GSE214480 and GSE214487, respectively. There is no restriction on data availability. No code was developed in this study. Further information and requests for resources and reagents should be directed to and will be fulfilled by the Lead Contact, Xingxing Zang (xingxing.zang@einsteinmed.edu). Source data are provided with this paper.

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

## Acknowledgements

We are grateful to our patients for their time and precious tissue samples. We thank Dr. Ulrich Steidl for sharing human AML cell lines and the Flow Cytometry Core Facility within the Einstein Cancer Center (funded by NIH P30CA013330) for the use of flow cytometers. This work was supported by National Institutes of Health (NIH) grants R01CA175495 (to X.Z.), R01CA262132 (to X.Z.), and R01HL152637 (to D.F.); Department of Defense PC210331 (to X.Z.); Irma T. Hirschl/Monique Weill-Caulier Trust (to X.Z.); Cancer Research Institute (to X.Z.); and Montefiore Einstein Cancer Center/Price Family Foundation (to XZ).

## Author contributions

H.W. and X.Z. developed the concept and designed the experiments; H.W. performed and analyzed the majority of the experiments.; R.A.S., G.K., C.N., X.R., A.T., and B.W. provided necessary clinical data, research materials, and advice; P.G. and D.Z. carried out RNA-seq analyses; Z.J. and D.F. conducted and analyzed intravital imaging; B.E.-G. provided human receptor interaction microarray screening; H.W. and X.Z. wrote the manuscript, and all other authors reviewed it.

## Competing interests

The authors declare the following competing interests. H.W., R.A.S., and X.Z. are inventors on a pending patent (Compositions and methods for inhibiting the expression of TMIGD2); B.E.-G. is an employee and X.Z. is the scientific co-founder of NextPoint Therapeutics. All other authors declare no competing interests.
