## [Peer Review File · Nature Communications]

REVIEWER COMMENTS

Reviewer #1 (Remarks to the Author):

In this manuscript, the authors provided substantial evidence about a functional role of TMIGD2, a T cell costimulatory receptor in AML. The authors did a nice work to characterize the functional role of TMIGD2 in AML biology, as well as providing with a detailed map of signaling cascades downstream of TMIGD2. Cell lines and primary AML samples are used for functional studies, both in vitro and vivo. Finally, the authors used antibodies to treat AMLs in PDX mouse model, demonstrating the therapeutic potential of targeting TMIGD2 in AML. Overall, it is a strong manuscript the reviewer favors to be published on Nat Commun.

Major comments:

1. AMLs at different differentiation stages can vary different, in terms of gene expression and therapeutic options. The expression of TMIGD2 in AML need to be elucidated in more details: is TMIGD2 expressed differently in different subtypes (M0-M5) of AMLs, or monocytic vs primitive? Any thought about what factor triggers the overexpression of TMIGD2 in LSCs over normal CD34+ stem cells?
2. The mechanism for the two TMIGD2 mAbs in anti-AML effect need to be better clarified: the authors have demonstrated that TMIGD2 provide with a beneficial signal for AML in a HHLA2-independent manner, and it also implies that TMIGD2 can forms a homodimer in vivo. The authors should test whether these two TMIGD2 antibodies disrupt the TMIGD2 cis-homodimerization.

Minor:

1. Data in Fig 1g does not provide with much valuable information.
2. The experiment in Fig.S6g won't distinguish trans vs cis interaction.
3. Will 20F2 treatment potentially delete human T cells, many of who express high level of TMIGD2? The author at least discuss this possible adverse effect with the usage of 20F2 in future application.

Reviewer #2 (Remarks to the Author):

Identifying new marker of leukemic stem cells (LSCs) and specifically targeting LSCs are the key to eliminate acute myeloid leukemia (AML) and have significant clinical importance. In this manuscript the authors describe that the transmembrane and immunoglobulin domain containing 2 (TMIGD2) is

aberrantly expressed in AML stem cells and is required for the expansion and maintenance of LSC self-renewal. Intriguingly, the authors show that TMIGD2 expresses in normal CD34+ HSPCs, which overlap with many CD34+ AML cells (Figure 1e) but has no significant role for normal hematopoiesis.

Mechanistically, the authors find TMIGD2 promotes LSC proliferation and blocks myeloid differentiation through the ERK1/2-p90RSK-CREB signaling axis. The authors further demonstrate that targeting TMIGD2 signaling with

anti-TMIGD2 monoclonal antibodies they generated dramatically attenuates LSC self-renewal and reduces leukemia burden in AML xenograft models. The data are robust in characterization of the phenotypes of TMIGD2 in AML, though lack of detailed mechanistic results, the topic is of interest to the reader of Nature Communication.

Major comments:

1. The molecule that relays the signaling from TMIGD2 to ERK1/2 is unclear. Is this related to Tyr192 on the TMIGD2 cytoplasmic tail?

2. In Figure 1a, the authors showed their gating strategy. However, a full gating strategy will be more informative. For example, in the right upper panel, please show the expression of HHLA2 and TMIGD2 in CD3- lymphocytes, which will provide some internal positive controls; please also check the expression of HHLA2 and TMIGD2 in monocyte gate (right lower panel). The authors use an AML in which almost all blasts are positive for CD34 as an example but beware of that not all CD34+ blasts are LSCs. Why the lineage markers only include CD14 and CD19, but not other markers including marker for NK cells as the blast gate by size and granularity is heterogeneous. On the right side, the majority of CD34+CD38+ and CD34+CD38- blasts are positive for TMIGD2. In my opinion, there are two populations in both CD34+CD38+ and CD34+CD38- fractions and the TMIGD2 gate should be moved further right. Note that the TMIGD2 gate is also different from Extended Data Figure 1e. Run normal cord blood and AML cells together will better appreciate the expression status of TMIGD2 in normal and leukemia cells. How do the authors gate AML with CD34- blasts? For example, in AML with monocytic differentiation or AML with NPM1 mutation. As the authors analyzed the expression of CD45RA, CD123 and IL1RAP (line 500), please show how these are gated and how they related to TMIGD2.

3. It is better to add CD34+CD38+TMIGD2+ and CD34+CD38+TMIGD2- cells in their colony assays in Figure 4b, which can compare the clonogenic capability of CD34+CD38-TMIGD2- vs CD34+CD38+TMIGD2+

4. Does overexpression of TMIGD2 in CD34+CD38-TMIGD2- cells enhance colony formation and in vivo engraftment of primary AML cells?

5. The authors compared the gene signatures of CD34+TMIGD2+ with CD34+TMIGD2- primary AML cells (Figure 4e-g), what are the gene signatures of TMIGD2^{hi} lin-CD34- AML cells in those AMLs with CD34- blasts?

6. In Figure 5, it is better to isolate CD34+CD38-TMIGD2+ and CD34+CD38-TMIGD2- cells first, then transduce with shCtrl or shTMIGD2 and perform engraftment experiments, which will tell if transduction of shTMIGD2 has any effect on CD34+CD38-TMIGD2- cells (negative control).

7. In Figure 6a-d, the authors use an erythroleukemia cell line HEL for mechanistic studies. As erythroleukemia is relatively rare among primary AMLs, it is better to confirm these results in primary human AMLs with CD34+ and CD34- blasts, respectively. In Figure 6h, what is the reason that the author switched to an acute monocytic leukemia cell line THP-1?

Related to the extended data Figure 6c, does overexpression of CREB rescue primary AML CD34+CD38-TMIGD2- cell proliferation, engraftment and further enhance CD34+CD38-TMIGD2+ cell function?

8. Related to Figure 7b-e, to strengthen the antibody's specificity, it is better to test the antibody effects using primary AML cells with or without TMIGD2 expression. Are the impacts of Abs on primary AML cells specific to LSCs or LSCs and bulk leukemic blasts? In Figure 7j, how many cells were transplanted for primary and secondary engraftment experiments? Do these cells (#31, #28, and #21) have higher phosphorylation of CREB? Does overexpression of CREB resist Ab treatment? What is the expression of TMIGD2 after Ab treatment? Without these data it is difficult to conclude that TMIGD2 served as a prognostic marker for quantification of minimal residual disease – line 357.

Minor comments:

1) In the Extended Data Figure 2b, the authors used erythroleukemia cell line HEL, CML blast crisis cell line K562, promyelocyte/macrophage cell line Kg1a, AML with RUNX1::RUNX1T1 cell line Kasumi1 and AML with inv(16) cell line ME-1 to investigate the expression of TMIGD2. As TMIGD2 expressed in almost all cells in HEL, Kasumi1 and ME-1, it will be difficult to explain its specific role in LSCs (the difference between cell lines and primary AML samples are noted, but the authors largely used AML cell lines for their studies, including mechanistic studies). Do these cell lines express HHLA2 and other TMIGD2 related coreceptors?

In addition, please show the quantification data of extended Data Figure 2f and 2h.

- 2) In Figure 2a, what is the knockdown efficiency of TMIGD2 and HHLA2 at mRNA level? Figure 2e, please show statistic results.

- 3) Please show the knockdown efficiency at the end of experiment in Figure 3f.

- 4) The font size in Figure 4e is too small to see. In Figure 4j, does knockdown of TMIGD2 affect T and NK cell chimerism (same for Figure 5h and Figure 7k)?

- 5) Please provide a table with summary of the used primary AML samples for colony assays and engraftment experiments.

- 6) Line 220, the authors stated that “TMIGD2 is essential for initiation and maintenance of leukemia”, what is the evidence that TMIGD2 is essential for leukemia initiation? The authors are using fully developed AML cell lines and primary AML cells.

- 7) In Extended Data Figure 7c, the authors gated all CD19- donor cells as CD33+. This cannot be correct.

- 8) Perform TMIGD2 immunostains on bone marrow core biopsy specimens from normal (e.g., negative lymphoma staging bone marrow) and AML samples will be more informative.

- 9) The authors did not show the number of experiments or biological replicates in many experiments, please check.

Point-by-Point Responses

The authors would like to express our sincere gratitude for giving us the chance to submit our revisited paper. We appreciate the editor and reviewers for taking time to carefully review the manuscript and give detailed and positive comments, which has greatly helped to improve this manuscript. It has taken us 10 months to finish additional experiments and address all comments. Below are our point-by-point responses to each comment of two reviewers. Original comments from the reviewers are in black font, and **author responses are in blue**.

The changes in the revised manuscript are **highlighted in yellow** for the convenience of the editor and the reviewers.

REVIEWER COMMENTS

Reviewer #1

In this manuscript, the authors provided substantial evidence about a functional role of TMIGD2, a T cell costimulatory receptor in AML. The authors did a nice work to characterize the functional role of TMIGD2 in AML biology, as well as providing with a detailed map of signaling cascades downstream of TMIGD2. Cell lines and primary AML samples are used for functional studies, both in vitro and vivo. Finally, the authors used antibodies to treat AMLs in PDX mouse model, demonstrating the therapeutic potential of targeting TMIGD2 in AML. Overall, it is a strong manuscript the reviewer favors to be published on Nat Commun.

We sincerely thank the reviewer for the positive evaluation of our study and his/her favoring our manuscript to be published on Nat Commun!

Major comments:

1. AMLs at different differentiation stages can vary different, in terms of gene expression and therapeutic options. The expression of TMIGD2 in AML need to be elucidated in more details: is TMIGD2 expressed differently in different subtypes (M0-M5) of AMLs, or monocytic vs primitive? Any thought about what factor triggers the overexpression of TMIGD2 in LSCs over normal CD34+ stem cells?

Response: We agree with the reviewer that these are important questions. To address the first question, we have analyzed the expression of TMIGD2 in AMLs at different differentiation stages using the AML TCGA database¹ (<http://www.cbioportal.org>). TMIGD2 was found to be more frequently expressed in FAB M0-M2 subtypes, acute promyelocytic leukemia (FAB M3), and FAB-M6 acute erythroid leukemia.

Modification in manuscript: Please see Page 3 and 4, line 65-67, and Supplementary Fig. 1d.

Response: Regarding the second point, the K562 ENCODE data predicts that the transcription factor RFX1 can bind the open chromatin region of TMIGD2² (as Fig. a shown below)

(<https://www.encodeproject.org/>). Also, based on the AML TCGA data, TMIGD2 expression is positively correlated with RFX1 expression at mRNA level¹ (as Fig. b shown below). Furthermore, RFX1 has been shown to be enriched in FLT3-ITD-specific subclones derived from

AML patients, which are able to expand *in vitro* and engraft *in vivo* with higher proliferative capacity than the FLT3-WT subclones³. We therefore hypothesize that RFX1 is potentially involved in the triggering of TMIGD2 overexpression in LSCs.

Modification in manuscript: Please see Page 12-13, line 358-363.

2. The mechanism for the two TMIGD2 mAbs in anti-AML effect need to be better clarified: the authors have demonstrated that TMIGD2 provide with a beneficial signal for AML in a HHLA2-independent manner, and it also implies that TMIGD2 can forms a homodimer *in vivo*. The authors should test whether these two TMIGD2 antibodies disrupt the TMIGD2 cis-homodimerization.

Response: The reviewer makes an excellent point. We have performed additional experiments to dissect the impact of these two TMIGD2 mAbs on disrupting the TMIGD2 cis-homodimerization.

Modification in manuscript: Please see Page 10, line 291-292, and Supplementary Fig. 7a.

Minor:

1. Data in Fig 1g does not provide with much valuable information.

Response: Fig 1g emphasizes that the TMIGD2 expression is significantly higher in leukemia cells than T cells in the same AML individuals. Due to repetitive neoantigen stimulation, TMIGD2 expression is decreased in T cells from AML patients. Therefore, it might be possible to target leukemia cells over T cells with anti-TMIGD2 mAb treatment (Please see also our response to minor comment #3).

2. The experiment in Fig.S6g won't distinguish trans vs cis interaction.

Response: If TMIGD2 can bind with TMIGD2 *in-trans*, we should be able to see binding signal by flow cytometry. Whereas, the TMIGD2-TMIGD2 cis interaction cannot be detected by flow cytometry. For example, B7-1 binds with PD-L1 in cis, which cannot be tested by flow cytometry⁴. The Fig.S6g revealed that there was no TMIGD2-TMIGD2 in-trans.

3. Will 20F2 treatment potentially delete human T cells, many of who express high level of TMIGD2? The author at least discuss this possible adverse effect with the usage of 20F2 in future application.

Response: We agree that the 20F2 treatment may delete human T cells via ADCP in the patient-derived xenograft mouse model. In response to the comment, we have now performed additional experiments by engrafting mononuclear cells, including immune cells and leukemia cells, from AML patients into irradiated NSG mice. Upon 20F2 treatment, we found that the percentage of hCD33+ leukemia cells was significantly decreased while the percentage of hCD3+ lymphocytes was variable but showed no significant difference. In order to advance 20F2 towards a future phase 1 clinical trial, we are developing humanized IgG4 version without ADCC/CDC/ADCP of 20F2.

Modification in manuscript: Please see Page 11, line 322, and Supplementary Fig. 7i.

Reviewer #2:

Identifying new marker of leukemic stem cells (LSCs) and specifically targeting LSCs are the key to eliminate acute myeloid leukemia (AML) and have significant clinical importance. In this manuscript the authors describe that the transmembrane and immunoglobulin domain containing 2 (TMIGD2) is aberrantly expressed in AML stem cells and is required for the expansion and maintenance of LSC self-renewal. Intriguingly, the authors show that TMIGD2 expresses in normal CD34+ HSPCs, which overlap with many CD34+ AML cells (Figure 1e) but has no significant role for normal hematopoiesis. Mechanistically, the authors find TMIGD2 promotes LSC proliferation and blocks myeloid differentiation through the ERK1/2-p90RSK-CREB signaling axis. The authors further demonstrate that targeting TMIGD2 signaling with anti-TMIGD2 monoclonal antibodies they generated dramatically attenuates LSC self-renewal

and reduces leukemia burden in AML xenograft models. The data are robust in characterization of the phenotypes of TMIGD2 in AML, though lack of detailed mechanistic results, the topic is of interest to the reader of Nature Communication.

We thank the reviewer for the constructive feedback that helped us to significantly improve our manuscript.

Major comments:

1. The molecule that relays the signaling from TMIGD2 to ERK1/2 is unclear. Is this related to Tyr192 on the TMIGD2 cytoplasmic tail?

Response: We agree that this is a valid point. It was previously reported that Tyr192, Tyr222 and Ser220, but not Tyr197 of TMIGD2, could potentially be phosphorylated, thereby mediating signal transduction^{5,6}. We attempted to examine the phosphorylation of Tyr192 in AML cell lines by performing immunoprecipitation, but we failed as our home-made and commercial anti-TMIGD2 mAbs did not work well for the assay. Alternatively, we performed flow cytometry using cells that express WT-TMIGD2 or TMIGD2 mutants upon the treatment with the CREB inhibitor, 666-15. We found that the TMIGD2 mediated signaling was mainly related to Tyr222.

Modification in manuscript: Please see Page 10, line 282-286, and Supplementary Fig. 6j.

2. In Figure 1a, the authors showed their gating strategy. However, a full gating strategy will be more informative. For example, in the right upper panel, please show the expression of HHLA2 and TMIGD2 in CD3⁻ lymphocytes, which will provide some internal positive controls; please also check the expression of HHLA2 and TMIGD2 in monocyte gate (right lower panel). The authors use an AML in which almost all blasts are positive for CD34 as an example but beware of that not all CD34⁺ blasts are LSCs. Why the lineage markers only include CD14 and CD19, but not other makers including marker for NK cells as the blast gate by size and granularity is heterogenous. On the right side, the majority of CD34⁺CD38⁺ and CD34⁺CD38⁻ blasts are positive for TMIGD2. In my opinion, there are two populations in both CD34⁺CD38⁺ and CD34⁺CD38⁻ fractions and the TMIGD2 gate should be move to further right. Note that the TMIGD2 gate is also different from Extended Data Figure 1e. Run normal cord blood and AML cells together will be better appreciate the expression status of TMIGD2 in normal and leukemia cells. How do the authors gate AML with CD34⁻ blasts? For example, in AML with monocytic differentiation or AML with NPM1 mutation. As the authors analyzed the expression of CD45RA, CD123 and IL1RAP (line 500), please show how these are gated and how they related to TMIGD2.

Response: We thank the reviewer for these suggestions and have edited the Fig. 1a accordingly. We agree that AML is a heterogenous disease and LSCs could reside in any subpopulations. Our results in Fig. 1 indicated that higher TMIGD2 expression was observed in LSC-residing populations.

We indeed had a NK cell marker CD56 in our staining panel as TMIGD2 is also expressed on NK cells. We decided to exclude NK cells by lymphocyte gating (CD45^{high}SSC^{low} subpopulation)

because: 1) most AML patient samples we collected had limited numbers of NK cells; 2) some CD34⁺ leukemia cells are also CD56⁺^{7,8}; and 3) CD56⁺ NK cells could be well separated by lymphocyte gating.

The TMIGD2 expression did not show distinct populations in most of AML samples. We therefore determined our HHLA2 and TMIGD2 gates by FMO control, which would be a good way to identify and gate positive populations in multicolor experiments.

In new Supplementary Fig. 1g (previous Extended Data Figure 1e), we performed flow cytometry using CD34⁺ cells enriched (Miltenyi CD34 MicroBead Kit) from normal bone marrow or cord blood. We tried mononuclear cells and found that it was hard to accurately quantify TMIGD2 expression due to too many CD34 negative but TMIGD2 positive cells.

We used the same gating strategy as shown in Fig. 1a (but focused on CD34⁻ blasts) to investigate TMIGD2 expression in AML with CD34⁻ blasts. We included a representative flow cytometry figure showing how these markers are gated and how they related to TMIGD2.

Modification in manuscript: Please see Page 3 and 4, line 60 and 74, Fig 1a and Supplementary Fig. 1e.

3. It is better to add CD34⁺CD38⁺TMIGD2⁺ and CD34⁺CD38⁺TMIGD2⁻ cells in their colony assays in Figure 4b, which can compare the clonogenic capability of CD34⁺CD38⁻TMIGD2⁻ vs CD34⁺CD38⁺TMIGD2⁺

Response: We have repeated this experiment and paid particular attention to the issues raised by the reviewer. The CD34⁺CD38⁺TMIGD2⁺ subpopulations showed much higher clonogenic capacity than CD34⁺CD38⁻TMIGD2⁻ subpopulations.

Modification in manuscript: Please see Page 7, line 166, and Supplementary Fig. 4b.

4. Does overexpression of TMIGD2 in CD34⁺CD38⁻TMIGD2⁻ cells enhance colony formation and in vivo engraftment of primary AML cells?

Response: We concerned that overexpressing TMIGD2 using lentivirus would drive expression of TMIGD2 to a high and non-physiological level. One possible way to physiologically upregulate the expression of TMIGD2 in primary AML cells is to perform electroporation with chemically-modified sgRNAs targeting the promoter region of TMIGD2 and catalytically inactive Cas9 fused to the tripartite activator VP64-p65-Rta (dCas9-VPR mRNA), which will take much more effort. In the original manuscript, we showed in multiple experiments that endogenous TMIGD2 knockdown affected primary human AML LSC capability (Fig 5).

5. The authors compared the gene signatures of CD34⁺TMIGD2⁺ with CD34⁺TMIGD2⁻ primary AML cells (Figure 4e-g), what are the gene signatures of TMIGD2[±] lin⁻CD34⁻ AML cells in those AMLs with CD34⁻ blasts?

Response: This is an important question. It has been reported that approximately 30% of cases of AML in which CD34 expression is low or absent^{9,10}, which is consistent with our samples. As

the Supplementary Table 2 shown, only 1/12 NPM1 mutant AML case (normally CD34⁺) expressed high TMIGD2. Considering this, the generation of gene signatures of TMIGD2[±] lin⁻ CD34⁻ AML cells in those AMLs with CD34⁺ blasts, which would be a good resource but would probably require additional 3+ years of work.

6. In Figure 5, it is better to isolate CD34⁺CD38⁻TMIGD2⁺ and CD34⁺CD38⁻TMIGD2⁻ cells first, then transduce with shCtrl or shTMIGD2 and perform engraftment experiments, which will tell if transduction of shTMIGD2 has any effect on CD34⁺CD38⁻TMIGD2⁻ cells (negative control).

Response: In these experiments, we wanted to study the functions of endogenous TMIGD2, so shCtrl were negative controls.

7. In Figure 6a-d, the authors use an erythroleukemia cell line HEL for mechanistic studies. As erythroleukemia is relatively rare among primary AMLs, it is better to confirm these results in primary human AMLs with CD34⁺ and CD34⁻ blasts, respectively. In Figure 6h, what is the reason that the author switched to an acute monocytic leukemia cell line THP-1?

Response: These results have been repeated using primary AML cells, please see Supplementary Fig. 6b. THP-1 is TMIGD2 negative, and it is widely used for AML research as a representative of M5 subtype. To not confine our results in certain subtypes of AML, we have tried to explore TMIGD2 function using different AML subtypes, including HEL (M6), Kasumi-1 (M2), Kg-1A (M4).

Related to the extended data Figure 6c, does overexpression of CREB rescue primary AML CD34⁺CD38⁻TMIGD2⁻ cell proliferation, engraftment and further enhance CD34⁺CD38⁻TMIGD2⁺ cell function?

Response: We agree that overexpression of CREB in primary CD34⁺CD38⁻TMIGD2^{+/-} AML cells would be an interesting experiment, but we would not expect that this is required. CREB has been found to enhance leukemogenesis in AML¹¹. The enhanced function by overexpression of CREB in TMIGD2^{+/-} primary cells might be unrelated with TMIGD2.

8. Related to Figure 7b-e, to strengthen the antibody's specificity, it is better to test the antibody effects using primary AML cells with or without TMIGD2 expression. Are the impacts of Abs on primary AML cells specific to LSCs or LSCs and bulk leukemic blasts? In Figure 7j, how many cells were transplanted for primary and secondary engraftment experiments? Do these cells (#31, #28, and #21) have higher phosphorylation of CREB? Does overexpression of CREB resist Ab treatment? What is the expression of TMIGD2 after Ab treatment? Without these data it is difficult to conclude that TMIGD2 served as a prognostic marker for quantification of minimal residual disease – line 357.

Response: Regarding antibody's specificity, we have assessed the antibody binding of three independent mAbs, including 17C7, 20F2, and a commercial clone (Clone # 953743) from R&D (as figure shown below). The three mAbs showed similar percentage of TMIGD2⁺ subpopulation in both T and NK cells from healthy PBMCs. Furthermore, we confirmed their specificity by staining TMIGD2 overexpressing (Supplementary Fig. 7b) and knock-down (Fig. 2a) cell lines.

Based on your suggestion, we have performed CFC assays using three primary AML samples without TMIGD2 expression. We found no significant different difference between isotype and 20F2 treatments (Supplementary Fig. 7d). We would not expect any therapeutic efficacy *in vivo* as TMIGD2 expression in leukemia cells remains negative. The majority of leukemic cells in AML patients are bulk leukemic blasts, and 20F2 could efficiently reduce leukemia burden in primary xenotransplantation within 2-3 weeks, suggesting that the mAb targeted bulk leukemic blasts. In addition, 20F2 treatment also targeted LSCs, as evidence by decreased percentage of CD34⁺CD38⁻ subpopulation *in vivo* (Supplementary Fig. 7k) and impaired capability of LSCs to initiate disease in the second round xenotransplantation (Fig. 7j).

We apologize for the confusion. More detailed information of the xenotransplantation experiments has been included in the revised manuscript (Fig. 7j). In Extended Data Fig. 6b, we validated the phospho-CREB level in control and TMIGD2-knockdown primary AML cells isolated from Pt#26 and Pt#28. The phospho-CREB level was variable among different AML patients. Regarding overexpression of CREB in primary AML cells, please also see Major Comment #4. Importantly, as a part of pre-clinical research, we would prefer to use primary AML cells without any further gene-editing to investigate the therapeutic efficacy of our anti-TMIGD2 mAbs. The expression of TMIGD2 after Ab treatment was shown in Supplementary Fig. 7j.

Modification in manuscript: Please see Page 11, line 295-297, and Supplementary Fig. 7d. Please see Page 12, line 328-330, and Supplementary Fig. 7j,k.

Minor comments:

1) In the Extended Data Figure 2b, the authors used erythroleukemia cell line HEL, CML blast

crisis cell line K562, promyelocyte/macrophage cell line Kg1a, AML with RUNX1::RUNX1T1 cell line Kasumi1 and AML with inv(16) cell line ME-1 to investigate the expression of TMIGD2. As TMIGD2 expressed in almost all cells in HEL, Kasumi1 and ME-1, it will be difficult to explain its specific role in LSCs (the difference between cell lines and primary AML samples are noted, but the authors largely used AML cell lines for their studies, including mechanistic studies). Do these cell lines express HHLA2 and other TMIGD2 related coreceptors?

Response: Thank you, we agree that this is an important question. We used TMIGD2 expressing cell lines as useful tools to elucidate the role of TMIGD2 in AML and the mechanism by which TMIGD2 regulates leukemia cells. We tried to make our experiments less artificial by performing multiple assays using primary AML samples and establishing PDX mouse models (Fig. 4, 5 and 7). We repeated the phenotypes found in Fig. 2 by primary leukemia cells as shown in Fig. 5b and 5c, and validated the downregulation of phospho-CREB in primary AML cells in Supplementary Fig. 6b.

HHLA2 has a co-stimulatory receptor TMIGD2 and a newly discovered co-inhibitory receptor killer cell Ig-like receptor, three Ig domains, and long cytoplasmic tail (KIR3DL3)¹². We have found limited expression of HHLA2 and KIR3DL3 on leukemia cells in patients with AML. Of note, HEL is the only cell line that express both HHLA2 and TMIGD2 although the expression of HHLA2 is low. KIR3DL3 expression was not observed in all these AML cell lines.

In addition, please show the quantification data of extended Data Figure 2f and 2h.

Response: Please see Fig. 2e, which is the quantification data for previous Supplementary Fig 2f. We have now shown quantification data for previous Supplementary Fig. 2h in revised manuscript.

Modification in manuscript: Please see Fig. 2e and Supplementary Fig. 2i.

2) In Figure 2a, what is the knockdown efficiency of TMIGD2 and HHLA2 at mRNA level? Figure 2e, please show statistic results.

Response: In response to the reviewer, we have performed additional experiments to validate the knockdown efficiency of TMIGD2 at mRNA level by qPCR (Supplementary Fig. 2c). We have also attempted to quantify HHLA2 in shCtrl and shHHLA2 HEL cells by qPCR, but the results were quite variable due to low HHLA2 mRNA levels in these HEL cells. Statistic results have been added to Fig. 2e in our revised manuscript.

Modification in manuscript: Please see Supplementary Fig. 2c and Fig. 2e.

3) Please show the knockdown efficiency at the end of experiment in Figure 3f.

Response: We thank the reviewer for this suggestion.

Modification in manuscript: Please see Supplementary Fig. 3c.

4) The font size in Figure 4e is too small to see. In Figure 4j, does knockdown of TMIGD2 affect T and NK cell chimerism (same for Figure 5h and Figure 7k)?

Response: Figure 4e has been edited as suggested by reviewer. Consistent with previous reports^{13,14}, we observed both CD19⁺ lymphocytes and CD33⁺ myeloid cells when normal CD34⁺ cells were engrafted (Supplementary Fig. 4j, 7l). There was almost no other cell types, including CD3⁺ T cells and CD56⁺ NK cells. We believe this is due to the xenograft mouse model we used. We transplanted human CD34⁺ cells into irradiated adult NSG mouse, which is unable to sufficiently support the development of human T and NK cells.

Modification in manuscript: Please see Fig. 4e.

5) Please provide a table with summary of the used primary AML samples for colony assays and engraftment experiments.

Response: Thank you for this suggestion. The summary of the used primary AML samples for colony assays and engraftment experiments has been attached as Supplementary Table 3.

6) Line 220, the authors stated that “TMIGD2 is essential for initiation and maintenance of leukemia”, what is the evidence that TMIGD2 is essential for leukemia initiation? The authors are using fully developed AML cell lines and primary AML cells.

Response: Thank you. We agree with reviewer’s point. We have replaced “TMIGD2 is essential for initiation and maintenance of leukemia” with “TMIGD2 is essential for the development of AML”.

Modification in manuscript: Please see Page 8, line 223.

7) In Extended Data Figure 7c, the authors gated all CD19⁻ donor cells as CD33⁺. This cannot be correct.

Response: We would like to clarify that for the PDX mouse model in previous Extended Data Figure 7c (now Supplementary Fig. 7e), we injected leukemia cells only, which resulted in predominantly myeloid engraftment (CD33 high/low). Please see also our response to Minor Comment #4.

8) Perform TMIGD2 immunostains on bone marrow core biopsy specimens from normal (e.g., negative lymphoma staging bone marrow) and AML samples will be more informative.

Response: This is an interesting point. In normal bone marrow specimens, the TMIGD2 expression might be relatively low in HSPCs while very high in plasmacytoid dendritic cells, innate lymphoid cells, T and NK cells; however, in bone marrow samples with AML, the TMIGD2 expression was upregulated in leukemia cells and downregulated in T and NK cells¹⁵. In addition to the expression level of TMIGD2 in different cell subsets, cellular components are also quite different in bone marrow biopsies from normal and AML samples. AML patients tend to have less T and NK cells, but more TMIGD2 expressing leukemia cells than healthy donors. Considering these information, it would be challenging to include all these markers when performing TMIGD2 immunostains. Besides, although our home-made anti-TMIGD2 mAbs were good for flow cytometry and *in vivo* therapy, they, and other commercially available TMIGD2 antibodies, didn’t work well for TMIGD2 immunostains. Therefore, in our work we investigated TMIGD2 expression largely by flow cytometry.

9) The authors did not show the number of experiments or biological replicates in many experiments, please check.

Response: In our revised manuscript we have shown the number of experiments or biological replicates.

References

1. Network, C.G.A.R. Genomic and epigenomic landscapes of adult de novo acute myeloid leukemia. *New England Journal of Medicine* **368**, 2059-2074 (2013).
2. Sloan, C.A., *et al.* ENCODE data at the ENCODE portal. *Nucleic acids research* **44**, D726-D732 (2016).
3. de Boer, B., *et al.* Prospective isolation and characterization of genetically and functionally distinct AML subclones. *Cancer Cell* **34**, 674-689. e678 (2018).
4. Chaudhri, A., *et al.* PD-L1 binds to B7-1 only in cis on the same cell surface. *Cancer immunology research* **6**, 921-929 (2018).
5. Zhu, Y., *et al.* B7-H5 costimulates human T cells via CD28H. *Nature communications* **4**, 2043 (2013).
6. Wang, Y.H.W., *et al.* IGPR-1 is required for endothelial cell–cell adhesion and barrier function. *Journal of molecular biology* **428**, 5019-5033 (2016).
7. Raspadori, D., *et al.* CD56 antigenic expression in acute myeloid leukemia identifies patients with poor clinical prognosis. *Leukemia* **15**, 1161-1164 (2001).
8. Coustan-Smith, E., Behm, F., Hurwitz, C., Rivera, G. & Campana, D. N-CAM (CD56) expression by CD34+ malignant myeloblasts has implications for minimal residual disease detection in acute myeloid leukemia. *Leukemia* **7**, 853-858 (1993).
9. Taussig, D.C., *et al.* Leukemia-initiating cells from some acute myeloid leukemia patients with mutated nucleophosmin reside in the CD34– fraction. *Blood* **115**, 1976-1984 (2010).
10. Quek, L., *et al.* Genetically distinct leukemic stem cells in human CD34– acute myeloid leukemia are arrested at a hemopoietic precursor-like stage. *Journal of Experimental Medicine* **213**, 1513-1535 (2016).
11. Shankar, D.B., *et al.* The role of CREB as a proto-oncogene in hematopoiesis and in acute myeloid leukemia. *Cancer Cell* **7**, 351-362 (2005).
12. Wei, Y., *et al.* KIR3DL3-HHLA2 is a human immunosuppressive pathway and a therapeutic target. *Science Immunology* **6**, eabf9792 (2021).
13. Chen, J., *et al.* Myelodysplastic syndrome progression to acute myeloid leukemia at the stem cell level. *Nature Medicine* **25**, 103-110 (2019).
14. Chung, S.S., *et al.* CD99 is a therapeutic target on disease stem cells in myeloid malignancies. *Science translational medicine* **9**, eaaj2025 (2017).

15. Crespo, J., *et al.* Phenotype and tissue distribution of CD28H+ immune cell subsets. *Oncoimmunology* **6**, e1362529 (2017).

REVIEWERS' COMMENTS

Reviewer #1 (Remarks to the Author):

The authors have addressed all my comments adequately. Thanks.

Reviewer #2 (Remarks to the Author):

I've noticed that the revisions made by the authors have significantly strengthened the manuscript. One aspect that may benefit from further development is not only identifying the critical downstream signaling cascades but also providing convincing evidence to validate their accuracy. Nevertheless, I genuinely appreciate all the hard work and dedication that the authors have put into revising the manuscript. Overall, I have no further concerns.

I would like to extend my best wishes to the authors for success in their future research endeavors.

Point-by-Point Responses

REVIEWER COMMENTS

Reviewer #1 (Remarks to the Author):

The authors have addressed all my comments adequately. Thanks.

Response: We sincerely thank the reviewer!

Reviewer #2 (Remarks to the Author):

I've noticed that the revisions made by the authors have significantly strengthened the manuscript. One aspect that may benefit from further development is not only identifying the critical downstream signaling cascades but also providing convincing evidence to validate their accuracy. Nevertheless, I genuinely appreciate all the hard work and dedication that the authors have put into revising the manuscript. Overall, I have no further concerns.

I would like to extend my best wishes to the authors for success in their future research endeavors.

Response: We sincerely thank the reviewer!